# Epigenetic loss of heterogeneity from low to high grade localized prostate tumours

Sebnem Ece Eksi [1,2 ✉], Alex Chitsazan[1], Zeynep Sayar[1,2], George V. Thomas [1,3], Andrew J. Fields[4], Ryan P. Kopp[5], Paul T. Spellman [1,4,6] & Andrew C. Adey [1,4,6 ✉]

Identifying precise molecular subtypes attributable to specific stages of localized prostate cancer has proven difficult due to high levels of heterogeneity. Bulk assays represent a population-average, which mask the heterogeneity that exists at the single-cell level. In this work, we sequence the accessible chromatin regions of 14,424 single-cells from 18 flash-frozen prostate tumours. We observe shared chromatin features among low-grade prostate cancer cells are lost in high-grade tumours. Despite this loss, high-grade tumours exhibit an enrichment for FOXA1, HOXB13 and CDX2 transcription factor binding sites, indicating a shared trans-regulatory programme. We identify two unique genes encoding neuronal adhesion molecules that are highly accessible in high-grade prostate tumours. We show NRXN1 and NLGN1 expression in epithelial, endothelial, immune and neuronal cells in prostate cancer using cyclic immunofluorescence. Our results provide a deeper under-standing of the active gene regulatory networks in primary prostate tumours, critical for molecular stratification of the disease.

[1] Cancer Early Detection Advanced Research (CEDAR), Knight Cancer Institute, OHSU, Portland, OR 97239, USA. [2] Department of Biomedical Engineering, School of Medicine, OHSU, Portland, OR 97209, USA. [3] Department of Pathology & Laboratory Medicine, School of Medicine, OHSU, Portland, OR 97239, USA. [4] Department of Molecular and Medical Genetics, School of Medicine, OHSU, Portland, OR 97239, USA. [5] Department of Urology, School of Medicine, OHSU, Portland, OR 97239, USA. [6] These authors jointly supervised this work: Paul T. Spellman, Andrew C. Adey. ✉email: eksi@ohsu.edu; adey@ohsu.edu

Tumour heterogeneity in prostate cancer poses a significant problem for molecular stratification of patients with localized prostate tumours[1,2]. It is well established that only a subset of clinically identified prostate cancers leads to lethal metastatic disease[3–5]. However, significant heterogeneity within a specific tumour focus and across different tumour foci in the prostate gland results in complex evolutionary trajectories for the disease[1,6–8]. The molecular heterogeneity within a tumour focus often leads to misclassification of the tumour grade and ineffective clinical treatment plans for many patients. The majority of the prostate cancer genomics and epigenomics data acquired to date originate from the bulk analysis of tumours that capture a population-average of different cell types in the tumour[9,10]. This generates a three-fold problem: (1) epithelial, endothelial, myeloid, lymphoid, nerve and other stromal cells that contribute to prostate cancer progression are reduced to a single component; (2) as a result, the dynamic bidirectional communication between these distinct components are not captured; (3) the heterogenous cell states within a single histopathological tumour grade are eliminated from the analysis.

Newly emergent single-cell technologies hold the key to profiling the vast heterogeneous landscapes of prostate cancer[11–17]. Recently, whole genomes of 20 single cells from localized prostate tumours were sequenced[18], revealing significant cell-to-cell variation in mutations and complex subclonal trajectories. However, these microdissection-based studies do not sample enough cancer cells to represent an unbiased image of localized prostate tumours. Combinatorial indexing of single-cells provides a way to profile thousands of cells from various types of tissues[19–21]. Currently, this method has been applied to a select group of cancers[22,23]. However, there are no current studies using these high-throughput single-cell technologies to characterize localized prostate adenocarcinoma.

To reveal the transformative changes in localized prostate tumours that lead to aggressive disease, it is important to capture the chromatin accessibility profiles of cells in low-grade and high-grade tumours. Open chromatin regions of cells contain not only promoter regions of actively transcribed genes, but also non-coding regulatory sequences. These sequences reflect the active gene regulatory networks that drive cell state transitions. Therefore, ATAC-seq (Assay for Transposase-Accessible Chromatin sequencing) technology provides a way to characterize both *cis*- and trans-regulators of cell states during tumour progression[24,25].

The current best predictor of outcome in localized prostate cancer is the degree of differentiation or grade of the tumour[26]. Grading of prostate cancer is reported through the Gleason score, which is a composite of the two most predominant Gleason grade patterns present in a sample[26]. Tumours that contain solely Gleason pattern 3 are often clinically indolent and tumours with higher Gleason pattern (≥4) are clinically significant and associated with a much worse outcome. This association remains constant even if the higher pattern tumour foci make up a small proportion of the entire tumour population[26]. It should also be noted that if followed long enough, some of the patients that are diagnosed with Gleason pattern 3 tumours also develop aggressive disease under active surveillance[4,27]. Therefore, determining the treatment strategies for patients with Gleason pattern 3 and 4 prostate tumours present a clinical challenge[27–29].

Understanding the transition between indolent and aggressive disease requires determining the risk of progression[30]. Even though it is important to properly stratify tumours based on the Gleason pattern and score, confounding factors exist, such as the surgical margin status for patients who have gone through radical prostatectomy surgery, presence or absence of extracapsular extension and lymph node involvement. The Cancer of the Prostate Risk Assessment Post-Surgical (CAPRA-S) score and

post-radical prostatectomy nomogram (Memorial Sloan Kettering Cancer Centre) take these additional factors into consideration and provide ways to delineate the probability of disease recurrence[31–34].

Our primary objective in this study is to identify molecular and cellular markers associated with prostate tumours that are primary Gleason pattern 3 (low-grade) and 4 (high-grade) and to explore the heterogeneity of the localized disease. Here we show, merging single-cell chromatin accessibility data with multiplex imaging, the *cis*- and trans-regulators of localized prostate tumours. We report that low-grade prostate tumours carry chromatin constraints, which are lost in high-grade tumours. We identify transcriptional factor binding signatures associated with low- and high-grade prostate tumours. Finally, we uncover the expression of neuronal adhesion molecules in prostate cancer and its surrounding stroma.

## Results

**Single-cell chromatin landscape of primary prostate tumours.**
We used combinatorial indexing to perform single-cell ATAC-seq (sci-ATAC-seq) from fresh-frozen prostate tumours collected from 18 patients via radical prostatectomy (Fig. 1a). FFPE tissue blocks were prepared from each patient, and both H&E sections and unstained adjacent sections for immunofluorescent (IF) staining were obtained from each block (Fig. 1a). Our cohort consists of six low-risk, nine intermediate-risk and two high-risk patients based on their CAPRA-S scores and Memorial Sloan Kettering Cancer Centre (MSKCC) nomograms (Supplementary Data 1)[31,33]. The majority of the tumours consist of primary Gleason pattern 3 (low-grade) and 4 (high-grade) (Supplementary Data 1).

We analyzed the distribution of captured open chromatin regions across different functional genomic elements such as the promoter, 3′UTR, 5′UTR, exon, intron and distal regions (Fig. 1b). We observed that aggregated sci-ATAC-seq data from all tumours exhibit a similar distribution across functional genomic elements when compared to bulk ATAC-seq results from the prostate adenocarcinoma (PRAD) TCGA data sets (Fig. 1b and Supplementary Fig. 1). This result indicates that sci-ATAC-seq is not biased in capturing open chromatin regions from specific genomic regions[35].

To confirm the quality of our acquired sci-ATAC-seq data sets, we aggregated peaks from Gleason pattern 3 and ≥4 tumours. We called 125,569 peaks and identified differentially accessible peaks around the promoter regions of several key genes in prostate cancer such as MYC and KLK3 (Fig. 1c). To further validate the quality of our sci-ATAC-seq data sets, we examined differential accessibility to the region that encodes for SCHLAP1 long non-coding RNA in tumours with Gleason pattern ≥4 as compared to pattern 3. SCHLAP1 is a prognostic marker that exhibits high expression in metastatic and high-grade prostate cancers[36,37]. In agreement with this, our results show higher accessibility to SCHLAP1 in Gleason pattern ≥4 tumours (Fig. 1c).

Following several quality control procedures (Supplementary Fig. 1, "Methods"), 14,424 cells were recovered with high-quality sci-ATAC-seq reads from 18 primary prostate cancer samples (Fig. 2a and Supplementary Fig. 1). snapATAC analysis of our sci-ATAC-seq data, using the Latent Dirichlet Allocation method, identified 16 clusters of cells[38,39]. Based upon cisTopic, each cluster of cells shares accessibility profiles grouped in 30 Topics (Fig. 2b)[40]. 16 clusters are visualized by the Uniform Manifold Approximation and Projection (UMAP) (Fig. 2a, b)[41,42]. A heatmap of topic scores across clusters shows distinct chromatin accessibility profiles of cells (Fig. 2c). Topics 4, 12 and 5 are mainly shared across cells from all clusters whereas Topics 20, 6

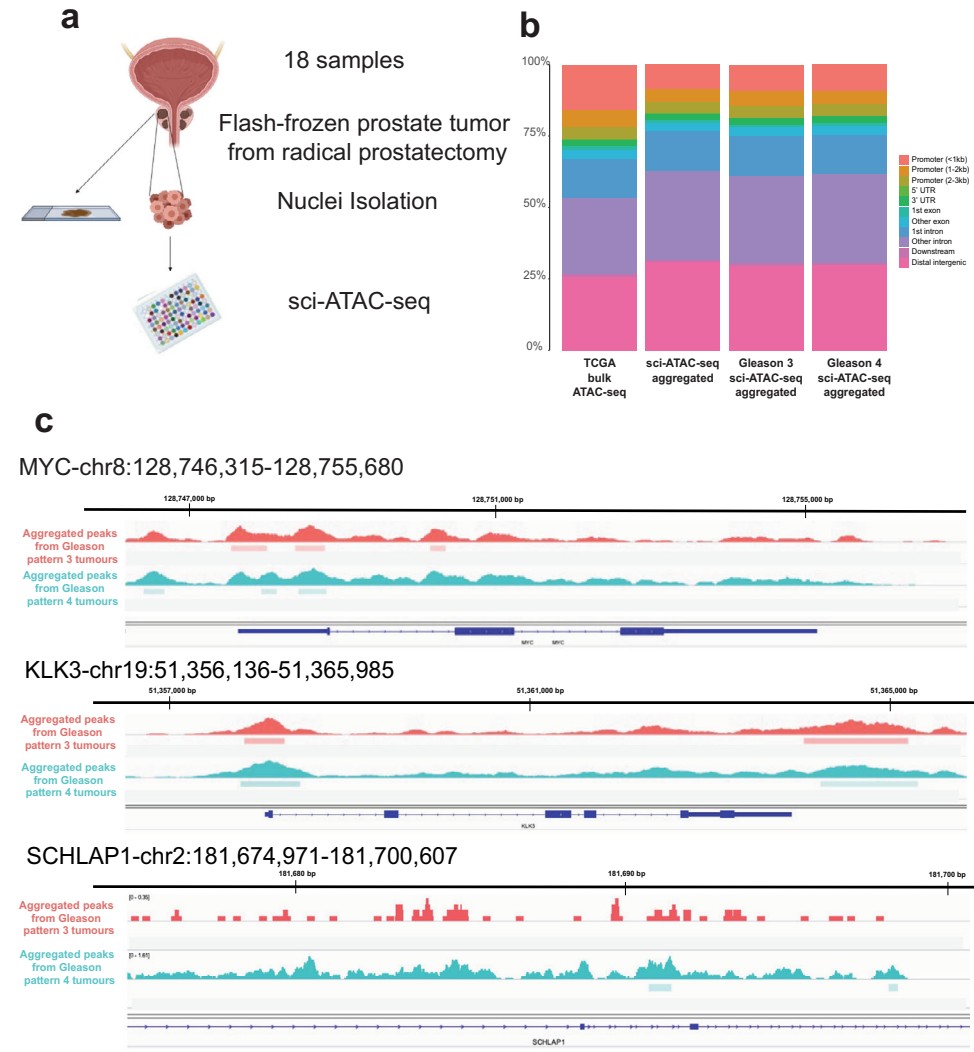

**Fig. 1 sci-ATAC-seq captures the chromatin accessibility landscape of prostate tumours with Gleason pattern 3 and 4. a** Diagram of experimental design. Flash-frozen prostate tumour samples were recovered from radical prostatectomies. Nuclei from each tumour were extracted and sorted into 96-well plates using unique combinatorial indices attached to transposase. FFPE tissue sections were collected from each patient along with an adjacent H&E section. Image created with BioRender.com. **b** Bar graph showing the percentage of peak distribution among functional genomic elements: promoter (<1 kb, 1—2 kb, and 2—3 kb), 3′UTR, 5′UTR, exon, intron, downstream (<300 kb) and distal regions. All aggregated peaks from Gleason pattern 3 and 4 tumours and TCGA bulk ATAC-seq results are shown. **c** sci-ATAC-seq peaks aggregated from Gleason pattern 3 (red) and 4 (blue) tumours. Called peaks are represented by dark blue rectangles. Examples of selected genomic regions: MYC gene (top), KLK3 (middle) and SCHLAP1 (bottom) are shown.

and 16 show very specific profiles for a small number of clusters (Supplementary Fig. 2a). We visualized putative gene transcription by quantifying the chromatin accessibility surrounding annotated transcription start sites (TSSs) (Fig. 2e).

**sci-ATAC-seq captures immune and stromal cell types.** Lymphoid, myeloid and other stromal cells have very different chromatin accessibility profiles compared to epithelial prostate cancer cells. Based on previous work, we expect that immune and stromal cell types associated with different prostate tumours would form distinct clusters of cells on the UMAP[43–45]. To examine this point, we used a cluster dendrogram of clusters to analyze the hierarchical relationship between the topics (Supplementary Fig. 2B). We observed Clusters 7, 12 and 14 attract cells from all tumour samples (Fig. 2a, b). To identify the accessible chromatin regions that define these Clusters, we analyzed all Topics using Genomic Regions Enrichment of Annotations Tool (GREAT) that takes a set of genomic regions as its input, finds the associated *cis*-regulatory regions and outputs annotation terms that are significantly enriched within the input genomic regions[46]. Topics 20 and 6 show an enrichment for GO terms related to the immune system process and immune response (Supplementary Fig. 2A). More specifically, immune Topic 20, which is enriched in cluster 14, contains GO terms associated with leukocyte and neutrophil activation, whereas immune Topic 6, which is enriched in cluster 12, contains GO terms associated with lymphocyte and T cell activation (Fig. 2c, d). Similarly, we observed an enrichment for genomic regions associated with extracellular matrix organization for stromal Topic 16, which is enriched in cluster 7 (Supplementary Fig. 2A). Altogether, our results show that cluster 7 consists of stromal cells, cluster 12 consists of lymphocytes and cluster 14 consists of myeloid immune cells. We eliminated clusters 7, 12 and 14 from our downstream analyses to identify the gene regulatory network changes that specifically occur in epithelial prostate cancer cells (Fig. 2d).

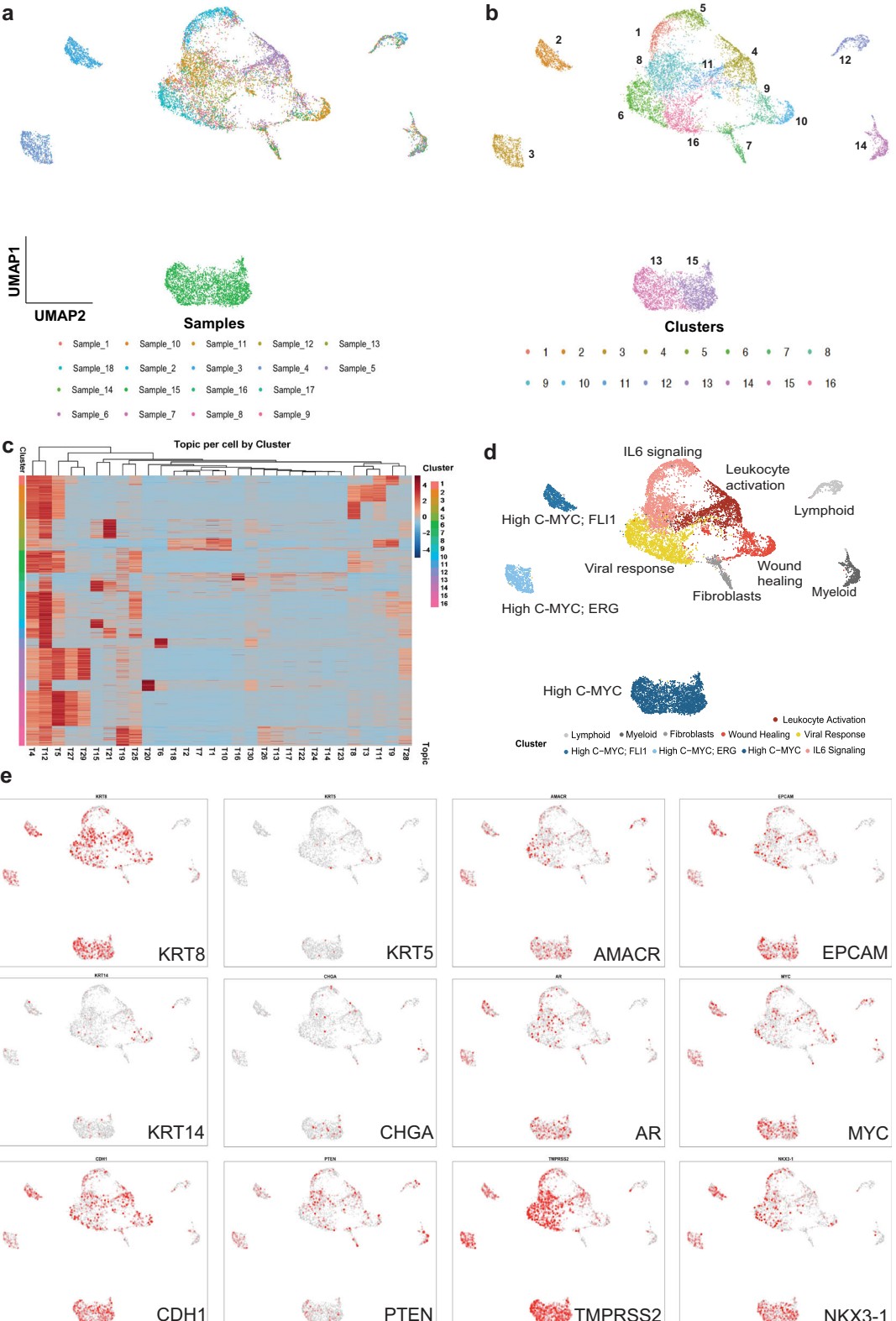

**Fig. 2 cisTOPIC identifies 16 clusters of cells spanning 30 topics from 14,251 cells. a** Single-cells from 18 prostate samples were clustered using cisTOPIC based on their chromatin accessibility profiles. Cells are coloured according to their sample IDs on the UMAP. **b** Single-cells from 18 prostate samples formed 16 clusters based on their shared chromatin accessibility profiles. Cells are coloured according to their cluster IDs on the UMAP. **c** Heatmap showing the topic distribution across 16 clusters. **d** Epithelial and stromal cell types are identified in localized prostate tumours. Lymphoid, myeloid and fibroblasts (grey tones) are removed from downstream analyses. Outer clusters show high-MYC accessibility (blue tones). The middle cluster shows higher accessibility to genes associated with inflammatory response (red tones). **e** Gene scores are shown for epithelial cell type markers luminal (KRT8), basal (KRT5 and KRT14) and neuroendocrine (Chromogranin A); common prostate epithelial cell markers AR, TMPRSS2 NKX3.1; prostate cancer cell markers AMACR, EPCAM, CDH1, C-MYC and PTEN.

**Epithelial prostate cancer cells carry markers of inflammatory response**. We used GREAT analysis in addition to the gene-based annotation tools of snapATAC to annotate the rest of the clusters identified through the topic analysis. We inferred the gene expression of marker genes for prostate tissue KRT8 (luminal epithelial), KRT5 (basal epithelial), KRT14 (basal epithelial), Chromogranin A (neuroendocrine), AR, TMPRSS2 and NKX3.1 (Fig. 2e). We also examined markers for prostate cancer such as AMACR, EPCAM and C-MYC as well as molecular markers CDH1 and PTEN (Fig. 2e). Our results show that the majority of our cells come from luminal epithelial cells as expected, with high accessibility to cancer markers AMACR and EPCAM (Fig. 2e). Neuroendocrine cells, which normally constitute a small percentage of the epithelial cell population, do not form independent clusters on the UMAP (Fig. 2e). Basal cells are observed in a small cluster defined by Topics 8 and 10 (Fig. 2e). GREAT analysis shows an enrichment for regions associated with early inflammatory response in cells that form the main cluster island on the UMAP. Topics 9, 15, 19 and 21 define the cells in the main cluster, which include gene terms related to IL-6 signalling, wound healing, viral response and leukocyte activation, respectively (Fig. 2d and Supplementary Fig. 3). We observed high accessibility for C-MYC for the three outer clusters (Fig. 2e). One of these outer clusters, also shows high accessibility for ERG and the other for FLI1 (Supplementary Fig. 7A). We decided to further delineate the biological differences between the main cluster and the three distant clusters of cells on the UMAP (Fig. 2e).

**Differentially accessible regions of Gleason pattern 4 prostate tumours**. To investigate the chromatin accessibility profiles of epithelial prostate cancer cells, we projected the clinical grades of the patient cohort on the UMAP (Supplementary Data 1 and Fig. 3a). We observe that the majority of the cells extracted from low- and intermediate-risk prostate patients (Gleason 3 + 3, Gleason 3 + 4) contribute to the single large cluster in the dimensionally reduced UMAP space (Fig. 3a and Supplementary Fig. 3). In contrast, single cells from one intermediate-risk (Gleason 4 + 4) and two high-risk (Gleason 4 + 4) prostate patients form the three distinct outer clusters (Fig. 3a and Supplementary Data 1). One intermediate-risk prostate tumour (Gleason 3 + 4), which consists of 70% Gleason pattern 3 cells and 30% Gleason pattern 4 cells, has cells mostly clustering with low-grade prostate tumours, and a small group of cells clustering with high-grade prostate tumours (Fig. 3a). Based on these results, we assigned single-cells to either Gleason pattern 3 or 4 categories on the UMAP (Fig. 3b), to examine the *cis*-regulatory and *trans*-regulatory differences observed in cells from Gleason pattern 3 (low-grade, main cluster) and Gleason pattern 4 (high-grade, outer clusters).

Differential accessibility functions of snapATAC ("Methods"[39],) identified top regions (Fig. 3c and Supplementary Figs. 4, 5) that are significantly accessible in Gleason pattern 4 tumours as compared to Gleason pattern 3 tumours. GREAT GO terms associated with accessible regions enriched in Gleason pattern 4 tumours include neuronal membrane adhesion, embryonic skeletal system morphogenesis, palate development and L-alanine transport (Fig. 3d). Notably, genomic regions associated with the neuronal adhesion genes, NRXN1, NLGN1 and CDH9, are highly accessible in Gleason pattern 4 vs. 3 tumours (Fig. 3d and Supplementary Fig. 5).

To determine if Gleason pattern 3 tumours would form separate clusters when analyzed in isolation, we eliminated all Gleason pattern 4 tumours from our analysis and performed topic analysis on Gleason pattern 3 tumours alone. We observed that Gleason pattern 3 tumours again formed one single cluster

and did not form patient-specific clusters, suggesting certain chromatin accessibility features are shared across all Gleason pattern 3 tumours (Supplementary Fig. 6).

We also profiled all cells based on their accessibility for known prostate cancer molecular subtypes (Supplementary Fig. 7)[47]. ERG, ETV1, ETV4 and FLI1 display patient-specific patterns even though accessibility of these markers do not drive clustering of cells within the data (Supplementary Fig. 7A). Samples 4 (Gleason score 4 + 4) and 16 (Gleason score 3 + 4) contain cells with high ERG accessibility, whereas the rest of the samples show heterogenous distribution for ERG accessibility (Supplementary Fig. 7A)[48]. Topics enriched in Samples 4 and 16 include genes such as AR, GRHL2, FOXA1, SLC43A1, WNT7B, which are known downstream players active in prostate cancers with ERG expression[49,50]. Sample 3 contain cells with high FLI1 accessibility. ETV1 and ETV4 show a heterogenous distribution (Supplementary Fig. 7A). We demonstrated the accessibility profiles for some of the recurrently mutated genes MYC, SPOP, PTEN and IDH1 across all our samples (Supplementary Fig. 7B). We observed higher accessibility to the promoter region of MYC in Gleason pattern 4 prostate tumours (Supplementary Fig. 7B).

**Gleason pattern 4 tumours are enriched for neuronal adhesion molecules**. Next, we used Cicero to examine the putative regulatory interactions around the NRXN1, NLGN1 and CDH9 loci[51]. Cicero is a single cell ATAC-seq method that finds putative regulatory interactions between regulatory sequences based on the co-accessibility of chromatin regions[51]. We observed an increase in the number of predicted *cis*-regulatory interactions around the NRXN1 locus in Gleason pattern 4 prostate tumours (Fig. 3e). Several regions distal to the NRXN1 promoter and intronic sequences within the gene body that are inaccessible in Gleason pattern 3 tumours, are accessible in Gleason pattern 4 tumours (Fig. 3e). Even though not quantitative, this result was particularly striking since we have a smaller number of Gleason pattern 4 cells (5,334 cells) than Gleason pattern 3 (7,383 cells). We also observed some additional putative regulatory interactions in Gleason pattern 4 tumours around the NLGN1 and CDH9 loci, even though the increase in the number of links were not as high as around the NRXN1 locus (Supplementary Fig. 7C).

To validate our results that show a significant enrichment in the chromatin accessibility profiles of high-grade tumours for neuronal adhesion molecules NRXN1, NLGN1 and CDH9, we analyzed the bulk ATAC-seq profiles using the TCGA PRAD data set[35]. TCGA PRAD data set consists of 26 patients with intermediate- and high-grade prostate tumours (Supplementary Fig. 8 and Supplementary Data 2). Only six patients have Gleason score 4 + 4 tumours and none of the patients have Gleason score 3 + 3 tumours. Despite these major clinical differences in the patient data sets, we detect a significant increase in the accessibility of NRXN1 and CDH9 chromatin sites in high-grade tumours as compared to intermediate-grade tumours (Supplementary Fig. 8). Next, we analyzed the bulk RNA-seq profiles of 497 patients from the TCGA cohort and detect NRXN1 expression in the majority of the prostate tumours (Supplementary Fig. 8). In contrast, we detect low levels of NLGN1 transcripts and no CDH9 transcription in the TCGA bulk RNA-seq data set (Supplementary Fig. 8).

**Loss of heterogeneity from Gleason pattern 3 to 4**. Our cluster analysis shows Gleason pattern 4 tumours as the outlier clusters on the UMAP whereas cells from primary Gleason pattern 3 tumours are mixed with each other in one cluster, regardless of patient ID. To further investigate the loss of heterogeneity from Gleason pattern 3 to 4 prostate tumours, we performed Silhouette

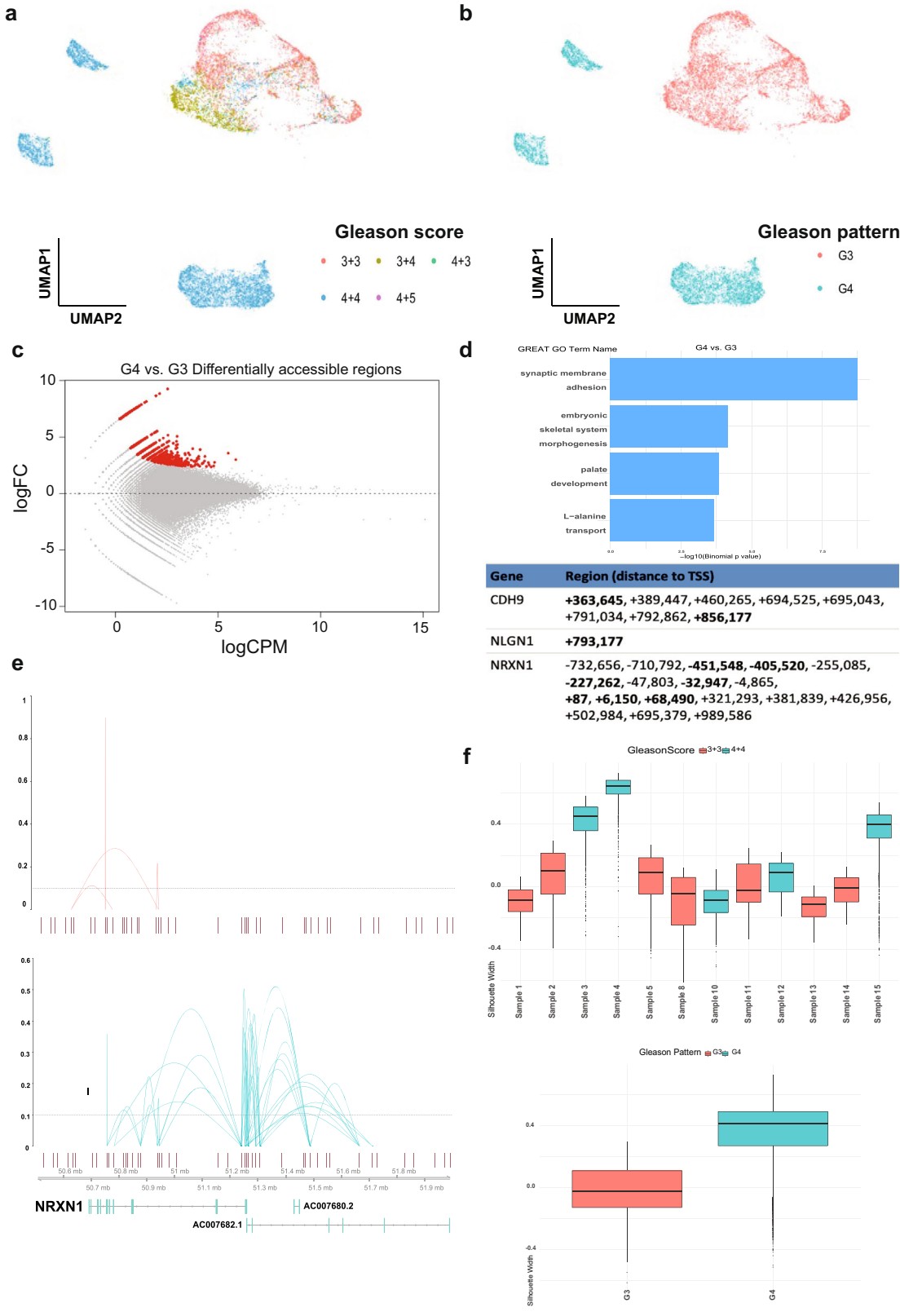

analysis to measure the distances between aggregated cells from Gleason pattern 3 and 4 tumours. Our results show cells from Gleason pattern 4 tumours to have decreased heterogeneity (Fig. 3f). To measure the level of heterogeneity within each primary Gleason pattern 3 and 4 samples, we also examined the Silhouette scores separately and found cells from Gleason pattern 3 tumours to consistently have higher heterogeneity as compared

to Gleason pattern 4 (Fig. 3f). To determine how this loss of heterogeneity is observed at the level of trans-regulators, we decided to examine the transcription factor (TF) binding motif analysis in Gleason pattern 3 and 4 prostate tumours.

**Distinct trans-regulatory networks of Gleason pattern 3 and 4 tumours.** sci-ATAC-seq gives a snapshot of all active genetic

**Fig. 3 Tumours with Gleason pattern 4 have distinct chromatin accessibility profiles compared to tumours with Gleason pattern 3. a** cisTOPIC-UMAP of 18 prostate samples coloured based on the Gleason score of each patient, which is a sum of two Gleason patterns: Gleason score 3 + 3 (red), Gleason score 3 + 4 (olive green), Gleason score 4 + 3 (green), Gleason score 4 + 4 (blue) and Gleason score 4 + 5 (magenta). **b** Gleason pattern 3 single-cells are red, while 4 are blue and assigned according to their cluster. **c** Distinct chromatin regions that are significantly more accessible in Gleason pattern 4 cells were identified (red). **d** Gene ontology results of the differentially accessible regions are shown (top). 15 peaks were linked to three neuronal adhesion genes, NRXN1, NLGN1 and CDH9. (+) sign indicates 3′ distal sequences, (−) sign indicates 5′ distal sequences to the transcription start site (TSS). **e** The putative interactions between the distal regulatory regions and promoter sequences of NRXN1 in Gleason pattern 3 (red) and Gleason pattern 4 (blue) tumours. Co-accessibility scores are shown on the y-axis and the dotted lines represent the threshold. Cicero links around the NRXN1 loci are significantly higher in number in Gleason pattern 4 vs 3 tumours. **f** Silhouette analysis using topic modelling for single-cells aggregated from each Gleason score 3 + 3 (red) and 4 + 4 (blue) prostate resolved by patient samples and all Gleason pattern 3 (red) and 4 (blue) prostate tumours aggregated. Sample 1 = 551 cells; Sample 2 = 796 cells; Sample 3 = 1087 cells; Sample 4 = 912 cells; Sample 5 = 1956 cells; Sample 6 = 95 cells; Sample 7 = 86 cells; Sample 8 = 122 cells; Sample 9 = 819 cells; Sample 10 = 387 cells; Sample 11 = 1582 cells; Sample 12 = 293 cells; Sample 13 = 439 cells; Sample 14 = 128 cells; Sample 15 = 3757 cells; Sample 16 = 244 cells; Sample 17 = 167 cells; Sample 18 = 821 cells. Quartiles are 25, 50 and 75% and 50% shows the median. The interquartile range is the difference between the 75th and 25th percentile. The upper whisker is the maximum value of the data that is within 1.5 times the interquartile range over the 75th percentile. The lower whisker is the minimum value of the data that is within 1.5 times the interquartile range over the 25th percentile. Outlier values are considered any values over 1.5 times the interquartile range over the 75th percentile or under the 25th percentile.

programmes in individual cells[19,52]. To understand the active gene regulatory networks in Gleason pattern 3 and 4 tumours, we aggregated all epithelial cells from Gleason pattern 3 and 4 tumours separately and analyzed the enriched TF binding motifs in each group using HOMER motif enrichment analysis[53]. We observe an enrichment for Fra1, Fra2, JunB, Atf3 and AP-1 in tumours with Gleason pattern 3 whereas tumours with Gleason pattern 4 show an enrichment for FOXA1, HOXB13 and CDX2 (Fig. 4a).

Next, to determine if different Gleason pattern 4 tumours from different patients share similar trans-regulatory networks, we analyzed the TF binding motifs of each Gleason pattern 4 tumour individually. We observe that each Gleason pattern 4 tumour is enriched for FOXA1, HOXB13 and CDX2 binding motifs, suggesting that higher-grade prostate tumours converge on the same trans-regulatory landscape (Fig. 4b). Specifically, we observe Sample_3 and Sample_4 to have a very similar TF binding motif enrichment profile, in terms of strong enrichment for FOXA1, HOXB13 and CDX2. Interestingly, Sample_15, a Gleason pattern 4 tumour, shows high enrichment for FOXA1, HOXB13 and CDX2, followed by Fra1/2, Atf3, JunB and AP-1 binding sites enriched in Gleason pattern 3 tumours, suggesting a transitionary state between the two regulatory states (Fig. 4b).

We observe increased accessibility for the promoter region of TFs HOXB13 and AR in prostate tumours with Gleason pattern 4 as compared to pattern 3 (Supplementary Fig. 9). FOXA1 is accessible across most prostate cancer cells and CDX2 shows higher accessibility, specifically in one of the Gleason pattern 4 tumours (Supplementary Fig. 9A). We also checked the patient outcome data using the Cistrome cancer database for TF profiles of FOXA1, HOXB13 and CDX2 in the TCGA PRAD data set[54,55]. We observe that patients with PRAD tumours have higher expression of FOXA1 and HOXB13 compared to the normal prostate tissue and poor survival is associated with high FOXA1 expression (Supplementary Fig. 10).

To validate the findings from our cohort, we compared the trans-regulatory landscapes of prostate tumours with different Gleason grades using the TCGA PRAD data set. In parallel to our observations, we found FOXA1, HOXB13 and CDX2 to be highly enriched in prostate tumours with primary Gleason pattern 4 (Gleason score 4 + 4 and 4 + 5 patients) as compared to tumours that are predominantly Gleason pattern 3 (Gleason score 3 + 4) (Fig. 4c). Similarly, we also observed Fra1/2, Atf3, JunB and AP-1 binding sites are enriched in predominantly Gleason pattern 3 tumours as compared to higher grade prostate tumours (Fig. 4c). When we analyzed the trans-regulatory differences between prostate tumours with a secondary Gleason pattern 5 (Gleason score 4 + 5) to tumours with Gleason pattern 4 (Gleason score 4 + 4) we find an enrichment of binding sites for class I steroid receptors, androgen (ARE), glucocorticoid (GRE) and progesterone (PG) (Fig. 4c). Interestingly, tumours with Gleason pattern 4 still show an enrichment for FOXA1, HOXB13 and CDX2, as well as for AP-1 like TFs as compared to tumours with a secondary Gleason pattern 5 (Fig. 4c).

**Epithelial, endothelial, immune and neuronal cells in prostate tumours express NRXN1 and NLGN1.** Neuronal adhesion molecules identified through our sci-ATAC-seq experiments, NRXN1 and NLGN1, are primarily expressed in the central nervous system and function in neuronal cell communication[56]. NRXN1 belongs to the family of neurexins, which is localized to the presynaptic membrane and interacts with neuroligins such as NLGN1, which is localized to the postsynaptic membrane[57]. The expression and function of NRXN1 and NLGN1 in prostate cancer have not been characterized before. Cyclic immunofluorescent (cyclic IF) microscopy provides in-depth information about molecular composition and spatial distribution of cellular heterogeneity by allowing the capture of more than 30 markers from single tissue sections[58,59]. To determine the expression pattern of NRXN1 and NLGN1 in prostate cancer across different cell types, we performed cyclic IF staining on tissue sections from eight patients in our cohort, using cell-type specific markers. To profile the spatial heterogeneity of NRXN1 and NLGN1 expression across distinct cell types, we marked basal (CK5, CK14), luminal (CK8) and neuroendocrine (Chromogranin A) epithelial cells within the prostate glands, as well as the endothelial (CD31), neuronal (NCAM) and immune cells (CD45, CD3) in the prostate cancer microenvironment (Fig. 5 and Supplementary Fig. 11). We validated antibodies against NRXN1 and NLGN1 using a brain tissue section as positive and colon tissue as negative control[60,61] (Supplementary Fig. 11).

We show that both NRXN1 and NLGN1 are expressed in basal, luminal and neuroendocrine cells in the prostate glands (Fig. 5d, e and Supplementary Fig. 12). We observe a slight increase in the number of neuroendocrine cells that express NRXN1 and NLGN1 in tissue sections from high-risk patients (Fig. 5d, e). We observe NRXN1 and NLGN1 are expressed in blood vessels in the prostate cancer tissue sections (Fig. 5a, d, e). Both our sci-ATAC-seq data and our cyclic IF results indicate NRXN1 and NLGN1 expression in immune cells infiltrating the prostate tumours (Supplementary Fig. 4A and Fig. 5d, e). Finally, we observe that both NRXN1 and NLGN1 are expressed in the N-CAM positive neuronal cells within the prostate tumour

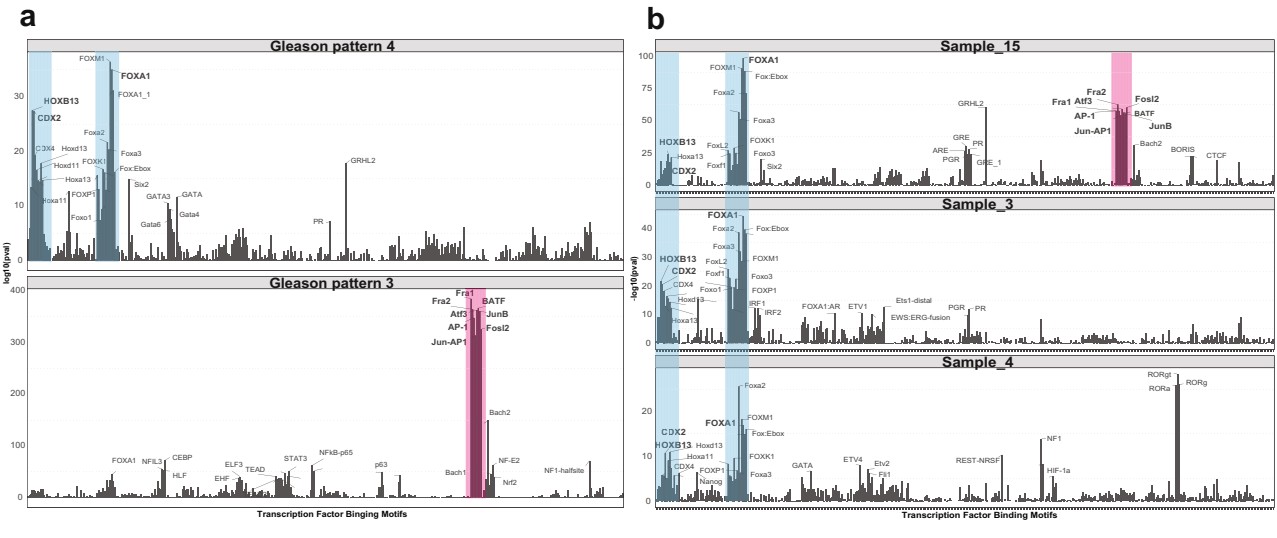

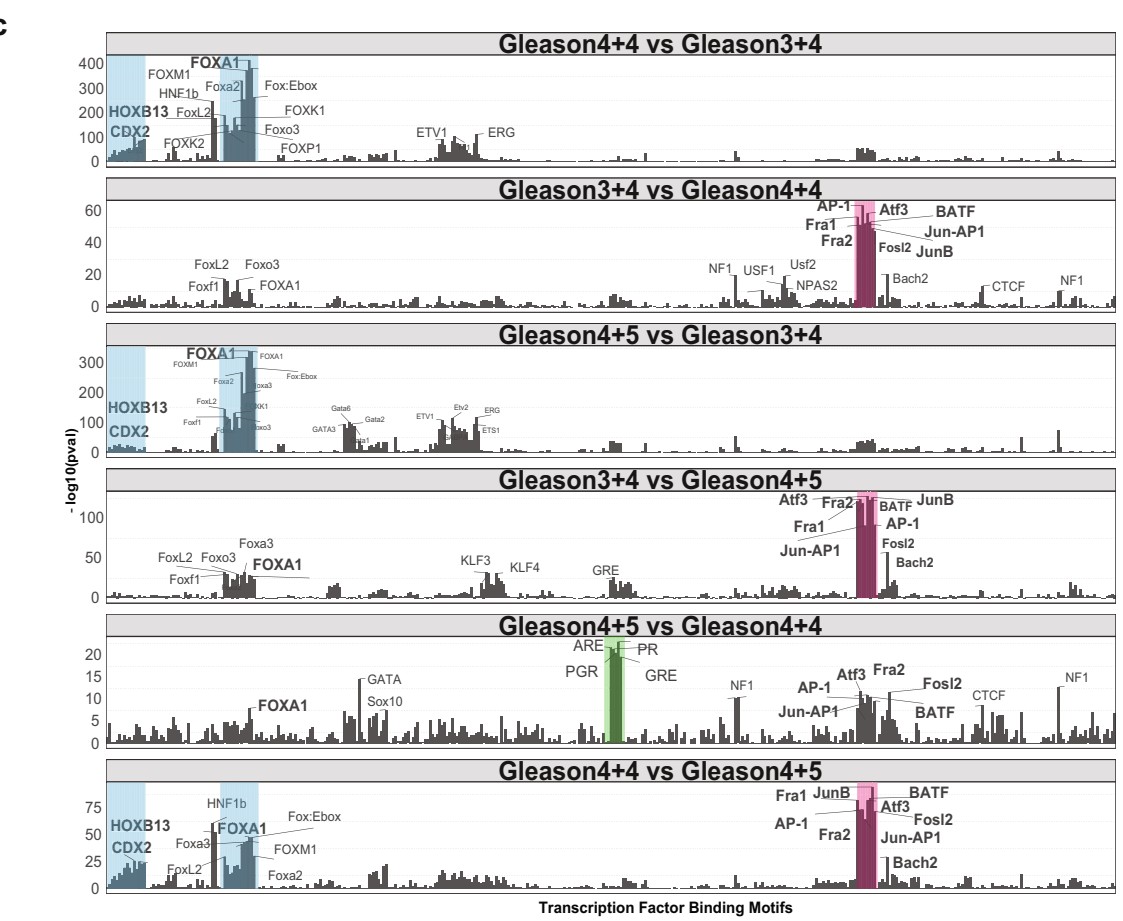

**Fig. 4 Common trans-regulatory networks exist in Gleason pattern 4 versus 3 prostate tumours. a** Transcription factor binding motifs enriched in aggregated single-cells from Gleason pattern 4 (top) and Gleason pattern 3 (bottom) tumours. Binding motifs are shown based on sequence similarities. Log scales vary due to differences in peak number and coverage. **b** Transcription factor binding motifs were enriched in aggregated single-cells from three different Gleason pattern 4 tumours. **c** Transcription factor binding motifs were enriched in prostate tumours from the TCGA cohort. Bulk ATAC-seq profiles from five Gleason score 3 + 4, six Gleason score 4 + 4 and ten Gleason score 4 + 5 patients were compared with each other.

stroma, with overall slightly higher expression profiles in high-risk patients (Fig. 5d, e).

## Discussion

Prostate cancer exhibits significant heterogeneity at the molecular, cellular and tissue level, hampering efforts to accurately

determine the risk of progression for localized prostate cancer. Single-cell technologies provide unique opportunities to accurately delineate the stages of prostate tumour heterogeneity prior to metastasis. However, single-cell technologies can also generate large data sets that may be difficult to validate and mechanistically resolve at the context of the tissue. Therefore, it is crucial

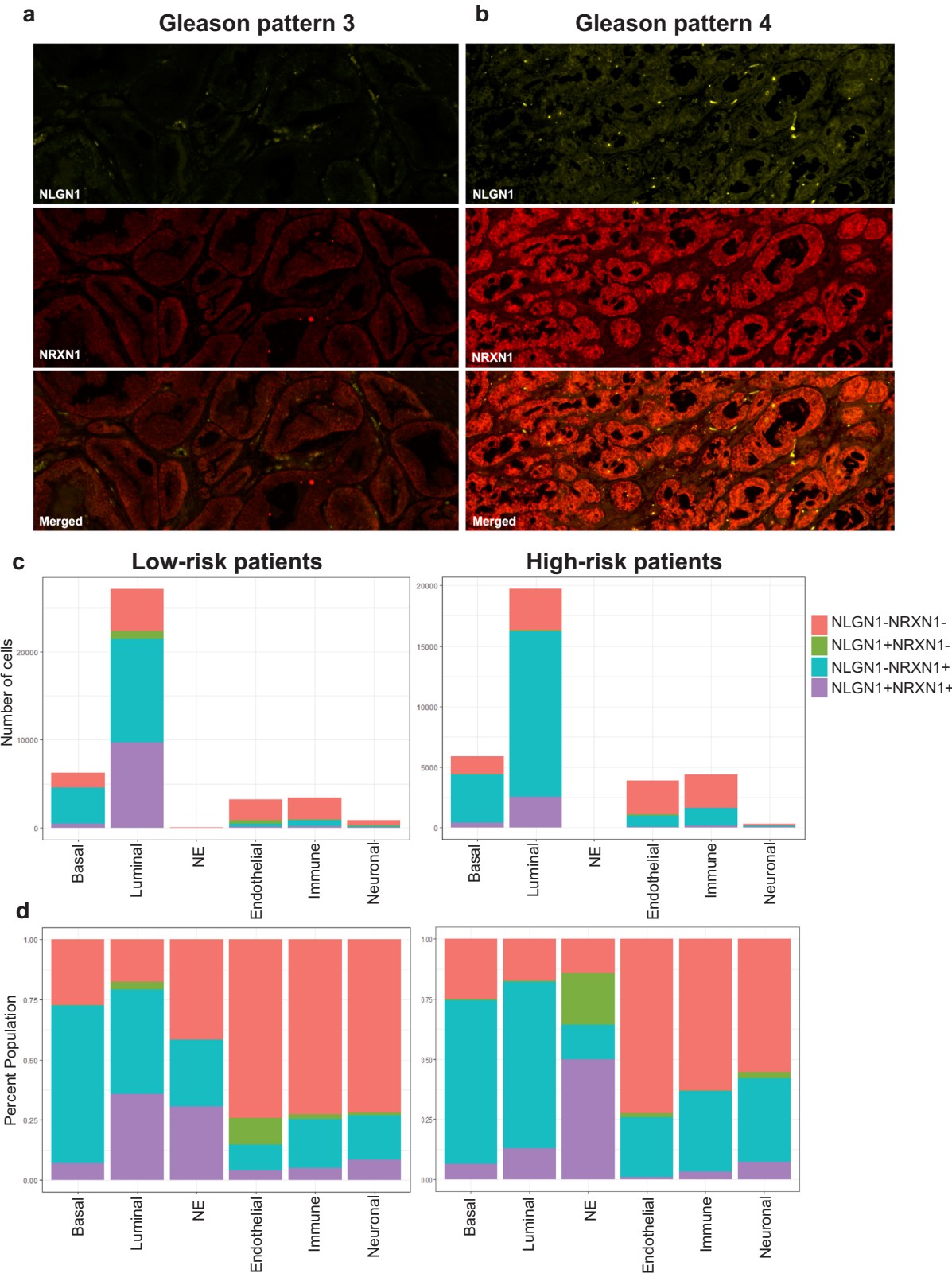

to merge these single-cell sequencing technologies with spatial imaging to provide an understanding of the transitionary states between indolent and aggressive cancers.

Using sci-ATAC-seq, we identified a co-accessibility pattern in neuronal adhesion molecules, NRXN1 and NLGN1, that distinguishes Gleason pattern 4 prostate tumours from Gleason pattern 3. Recent evidence shows that both the peripheral nervous system

and progenitors from the central nervous system may influence prostate cancer progression[62–64]. Neuronal cells within the prostate cancer microenvironment can release neurotransmitters that may modulate the behaviour of prostate cancer cells[65,66]. It has been shown that non-neuronal cells expressing neurexins and neuroligins result in pre- and post-synaptic specialization in neurons and these calcium-dependent synaptic molecules can

**Fig. 5 Neuronal adhesion molecules NRXN1 and NLGN1 are expressed in the epithelial, endothelial, immune and neuronal cells in prostate cancer.**
Protein expression of NLGN1 (yellow) and NRXN1 (red) in tumours with Gleason pattern 3, characterized by well-formed discrete glands with wide lumens (Sample_8) (**a**) and Gleason pattern 4, characterized by irregular glands with cribriform and branching architecture (Sample_4) (**b**). Supplementary Figure 10 shows H&E images and ROIs for each sample. Scale bars (white line) show 200 pixels. **c** Expression of NRXN1 and NLGN1 across different cell types from three tissue sections acquired from high-risk patients with Gleason pattern 4 prostate tumours in our cohort. X-axis shows cell types and Y-axis shows the total number of segmented cells (top) or percentage population of cells positive for that marker (bottom). NRXN1 expressing cells are shown in blue and NLGN1 in green. Cells that express both proteins are shown in purple and cells that do not express either are marked with pink. **d** Expression of NRXN1 and NLGN1 across different cell types from five tissue sections acquired from low-risk patients with Gleason pattern 3 tumours in our cohort. X-axis shows cell types and Y-axis shows the total number of segmented cells (top) or percentage population of cells positive for that marker (bottom). NRXN1 expressing cells are shown in blue and NLGN1 in green. Cells that express both proteins are shown in purple and cells that do not express either are shown in pink.

exploit calcium channels present in surrounding cells for their biological activity[67–69]. More interestingly, NLGN1 has shown to be cleaved enzymatically at its N-terminal ectodomain and secreted at excitatory synapses by enzymatic cleavage[70,71]. Similarly, NLGN3 has been shown to be secreted in an activity-dependent manner in glioma, promoting cell division and glioma growth[72]. Our cyclic IF analysis revealed the spatial distribution of NRXN1 and NLGN1 in prostate cancer and identified cell-type specific expression patterns for both proteins. We observed prostate epithelial cells, as well as immune and neuronal cells express these synaptic molecules in prostate cancer. Based on our findings, we propose that the expression of neuronal adhesion molecules in prostate cancer cells mark tumours for a more aggressive, potentially metastatic phenotype. Our findings suggest the possibility that NRXN1 and NLGN1 expression and secretion in high-grade prostate tumours may modulate the activity between cancer cells and neurons, which may contribute to perineural invasion or neoneurogenesis in prostate cancer[64,73,74].

Our study has provided an atlas of the regulatory landscape of low-grade (Gleason pattern 3) and high-grade (Gleason pattern 4) prostate tumours. Interestingly, we did not observe regulatory heterogeneity among single-cells from Gleason pattern 4 tumours. This may be because aggressive prostate tumours acquire distinct evolutionary trajectories that involve different types of chromosomal arrangements[75–77]. For instance, we observed at least one of the Gleason pattern 4 tumours had high levels of accessibility for the ERG promoter that led to a very distinct chromatin accessibility landscape for cells from this tumour. On the other hand, Gleason pattern 3 tumours exhibited significant cell-to-cell heterogeneity that led to clusters of cells from multiple Gleason pattern 3 tumours. Interestingly, all cells from Gleason pattern 3 tumours were bound by chromatin restraints that were lost in Gleason pattern 4 tumours. Further genomic studies are necessary to provide links between the common chromosomal rearrangements and single-cell epigenomic states of prostate cancer cells.

Our results also indicate unique trans-regulatory signatures for different grade localized prostate tumours. We found low-grade prostate tumours were significantly enriched for the AP-1 family of TF binding sites (JUN, JUNB, JUND and FOS, FOSB and FRA1, and the closely related activating TFs: ATF and CREB), in contrast to high-grade prostate tumours that were enriched for FOXA1, HOXB13 and CDX2. Interestingly, we observed at least one of the high-grade prostate tumours possessed signatures for both transcriptional regulatory programmes. We do not know whether this tumour represents a case that is undergoing a transition in chromatin structure, i.e., the low-grade prostate tumour regulatory marks are still present even though the tumour transitioned to a more aggressive state.

It is also important to point out that AP-1 family of TFs are responsible for the early inflammatory response in cancer and the top GO GREAT terms enriched in Gleason pattern 3 tumours are all related to inflammation. In other words, both our cis- and trans-regulatory analyses independently show that low-grade prostate tumours carry markers of inflammatory response, which supports previous studies that show a correlation between inflammatory markers and low-grade prostate cancer[78,79].

We also observed that all Gleason pattern 4 tumours in our cohort share the same trans-regulatory circuit, even though they formed distinct clusters of cells based on their chromatin accessibility profiles. Strikingly, we were able to identify the same trans-regulatory signature in Gleason pattern 4 and higher tumours in the TCGA data set. Our results, combined with the analysis of the TCGA data set, show that prostate tumours with Gleason pattern 4 share a common transcriptional regulatory programme defined by an enrichment of FOXA1, HOXB13 and CDX2 binding sites. This finding is particularly interesting since prostate cancers usually have low numbers of recurrent mutations[47,76,80]. Future research will determine whether disparate genomic changes observed in prostate cancer converge on common epigenomic profiles and if these present druggable targets in non-metastatic tumours.

Identification of gene regulatory programmes shared by high- and low-grade tumours present a strong opportunity to identify biomarkers for patient stratification despite the overwhelming molecular and cellular heterogeneity that exists in prostate cancer. There are several drugs in development targeting FOXA1 and HOXB13[81–83]. Our results suggest that these therapies could potentially benefit patients with high-grade non-metastatic prostate tumours. However, further mechanistic studies are required to better evaluate the potential effect of these drugs on chromatin structure and transcriptional regulation in localized tumours.

It is unclear whether the enriched binding sites for FOXA1, HOXB13 and CDX2 are dependent or independent of AR activity in single cells from high-grade tumours. It is known that FOXA1, HOXB13 and GATA2 act as pioneer TFs in the prostate tissue to facilitate androgen receptor (AR) transcription during prostate carcinogenesis[84]. However, their AR-independent activity has not been studied extensively. We anticipate that there is a similar synergy between FOXA1, HOXB13 and CDX2 TFs in remodelling the chromatin structure in high-grade prostate tumours. Previous studies show colocalization of TFs FOXA1 and HOXB13 at the reprogrammed AR binding sites in human prostate cancer cells and draw a link between TMPRSS2-ERG fusion and co-option of FOXA1 and HOXB13 to specific regulatory elements across the genome[85]. Interestingly, we observed an enrichment for FOXA1 and HOXB13 in Gleason pattern 4 tumours independent of their ERG accessibility (Supplementary Fig. 10). However, it is not possible to infer any details about the cooperation of these TFs with each other, their synergy with AR, and their functional impact on transcription based on our results. Additional characterization of histone modification markers is necessary to understand the details of the synergy between these

TFs on downstream gene expression in Gleason pattern 4 vs 3 prostate tumours.

## Methods

Our research complies with all relevant ethical regulations of Knight Cancer Institute at Oregon Health & Science University. Samples from patients undergoing radical prostatectomy were collected via informed consent through: (1) OHSU Knight Cancer Institute (KCI) Biolibrary under IRB#4918 and (2) Dr. Ryan Kopp's prostate cancer MRI study under IRB#18321. Participants were not compensated.

**Isolation of prostate cancer cells from radical prostatectomy samples**. Radical prostatectomy samples from 18 patients were obtained from the Biolibrary at OHSU. Fresh-frozen prostate samples positive for prostate adenocarcinoma were recovered from each radical prostatectomy specimen. H&E images were reviewed by a pathologist to confirm the Gleason grade and score of each sample, as well as the score assigned by the Biolibrary. Five unstained adjacent 5-micron tissue sections were obtained from each sample for immunofluorescent microscopy studies.

**Risk stratification of the patient cohort**. The Cancer of the Prostate Risk Assessment Post-Surgical (CAPRA-S) score and post-radical prostatectomy nomogram from Memorial Sloan Kettering Cancer Centre (MSKCC) were used to identify patients with low-, intermediate- and high-risk prostate cancer[31,34]. CAPRA-S score is calculated based on factors such as pathological Gleason score, surgical margin status and presence or absence of extracapsular extension and lymph node involvement for patients who have gone through radical prostatectomy surgery to estimate the probability of disease recurrence[32]. These factors were used to determine the specific clinical features of our patient cohort in comparison to the TCGA's cohort.

We also analyzed bulk ATAC-seq profiles of 21 patients available through the TCGA data set. We calculated the CAPRA scores for these patients and found that the majority of them were either intermediate- or high-risk as opposed to our cohort of patients that consisted mainly of low- or high-risk patients based on CAPRA risk stratification (Supplementary Data 2).

**Whole-mount sample acquisition**. Radical prostatectomy specimens were obtained via standard of care surgery and processed in the OHSU Pathology lab. Subjects consented to study OHSU IRB #18321. Under this protocol, an annotated radiology worksheet depicting the location of the MRI-detected tumour foci were completed by the radiologist, and a copy was sent with the specimen to the pathologist and grossing technician. Fresh prostate specimens were inked and sectioned at 5 mm intervals using a prostate slicing device (Procut P/5, Milestone Medical), followed by a gross examination to correlate gross findings with the annotated radiology worksheet. Fresh tissue was collected from tumour area with a 5 mm punch biopsy and removed for research purposes. A frozen section from this research tissue sample was used to confirm the presence of tumour. The entire remaining prostate was then submitted and processed for whole mount histology. Formalin fixed paraffin embedded (FFPE) tissue blocks and slides for standard clinical annotation were generated. This work was done under the supervision of a subspecialized genitourinary pathologist.

### sci-ATAC-seq sample processing

*Sample preparation and nuclei isolation*. The nuclei isolation protocol was improved and barcode space was extended to increase the multiplexing ability of the combinatorial indexing protocol. Frozen prostate tissue samples (0.1–0.8 g) were homogenized in Nuclei Isolation Buffer (NIB, 10 mM TrisHCl pH7.4, 10 mM NaCl, 3 mM MgCl2, 0.1% Igepal, 1 protease inhibitor tablet (Roche, Cat. 11873580001)) using a dounce homogenizer. Isolated nuclei were washed three times with ice cold 1XPBS and centrifuged down at $500 \times g$ for 5 min at 4 °C. Washed nuclei were passed through a 35 μm cell strainer (Corning) and stained with 5 μL (5 mg/ml) DAPI to mark the nuclei.

*Tn5 transposome assembly*. Tn5 enzyme was purified and loaded with specific oligo sequences[86]. Tn5 adaptor sequences synthesized at Integrated DNA technologies (Supplementary Data 3). Briefly, oligonucleotides for ME-rev (phosphorylated 19-basepair mosaic end) and i5 or i7 were incubated in equimolar amounts (100 μM each) for 5 min at 95 °C and cooled down slowly on the thermocycler in 3 °C increments. 0.25vol adaptor sequences, 0.4 vol 100% glycerol, 0.12 vol 2X dialysis buffer (100 mM HEPES–KOH at pH 7.2, 0.2 M NaCl, 0.2 mM EDTA, 2 mM DTT, 0.2% Triton X-100, 20% glycerol), 0.1 vol Tn5 (50 μM), 0.13 vol water. Uniquely indexed transposomes were stored in 96-well plates at −20 °C.

*Cell sorting*. For the first sort plate, 3000 DAPI stained nuclei were sorted into 96-well plates using the BD FacsAria Fusion cell sorter (FACSDiva v8.0.3). Each well contained 10 μL tagmentation buffer (5 μL NIB and 5 μL TD buffer from Illumina). For the second sort plate, 22–25 DAPI stained nuclei were sorted into 96-well plates using FacsAria cell sorter. Each well contained 8.5 μL master mix (0.25 μL 20 mg/ml BSA, 0.5 μL 1% SDS, 7.5 μL distilled water and 2.5 μL i5 and i7 PCR index primers).

*Tagmentation*. Tagmentation reaction was carried out at 55 °C for 30 min after the 1st nuclei sort. Once the reaction was at room temperature the plates were placed on ice and samples from each well were pooled in an Eppendorf tube. Pooled nuclei were passed through a 35 μm cell strainer (Corning) and stained with 5 μL (5 mg/ml) DAPI before the second nuclei sort. Samples were incubated at 55 °C for 15 min to denature the transposase after the second sort.

*PCR indexing*. Nuclei were amplified using RT-PCR (QuantStudio v1.7.1) for 20–25 cycles to insert unique i5-i7 DNA oligo sequences in each well based on the previously published protocol[52] (Supplementary Data 3). 7.5 μL NPM PCR mix (Illumina), 4 μL distilled water, 0.5 μL 100X SYBR Green dye was added to each well and the following PCR amplification cycle was followed: 75 °C for 5 min, 98 °C for 30 s, (for 22–25 cycles) 98 °C for 10 s, 63 °C for 30 s, 72 °C for 60 s, plate read at 72 °C for 10 s.

*Sample purification and fragment analysis*. 5 μL of sample from each well was pooled from each well and the library pool was purified using Qiagen PCR purification kit followed by AMPure bead purification. Each library pool was analyzed using Bioanalyzer to assess the quantity and distribution of fragment size before sequencing. Libraries were sequenced on the Next-seq platform (Illumina, Next-Seq500 NCS v4.0) using a 150-cycle kit with a custom sequencing recipe (Read 1: 47 imaged cycles; Index 1: 8 imaged cycles, 27 non-imaged/dark cycles, 10 imaged cycles; Index 2: 8 imaged cycles, 21 non-imaged / dark cycles, 10 imaged cycles; Read 2: 47 imaged cycles)[20].

**Immunohistochemistry**. Each formalin fixed and paraffin embedded (FFPE) prostate tissue block was serially sectioned. One Hematoxylin-and-Eosin (H&E) stained section and five adjacent unstained sections with 5-micron thickness were acquired from all tumours. FFPE tissue sections were deparaffinized as follows[58]: each tissue section was incubated in a 65 °C oven for 1 h and then immediately transferred in Xylene solution. Slides were sequentially immersed in fresh Xylene solution two times, 5 min each; 100% ethanol two times 5 min each; 95% ethanol two times 2 min each; 70% ethanol two times 2 min each and distilled water two times 5 min each. Antigen retrieval was done in a medical decloaking chamber filled with 0.5 L distilled water. Tissue slides were placed in a plastic jar that contains Citrate buffer (pH = 6) and incubated at high pressure for 15 min. Each slide was then dipped in distilled water and incubated in pH9 buffer for 15 min before being transferred to distilled water at room temperature to complete antigen retrieval. Slides were blocked with 10% NGS, 1%BSA in PBS for 30 min at room temperature inside a humidity chamber. Primary antibodies conjugated to Alexa Fluor dyes were prepared in 5% NGS, 1%BSA in PBS buffer. Conjugated primary antibodies used in this study are shown in Supplementary Fig. 9. All slides were imaged using Zeiss Axioscan.Z1 microscope (Zen2) located at the Knight Cancer Institute Research Building.

**Cyclic immunofluorescence microscopy**. For each round of immunofluorescent staining primary antibodies conjugated to Alexa Fluor dyes 488, 555, 647 and 750 were mixed in 1% BSA, 5% NGS solution using the following dilutions: NRXN1, Millipore Sigma, Concentration (1:100), Cat# ABN161-I; RRID:AB_11211973; NLGN1, Millipore Sigma, Concentration (1:100), Cat# MABN742; CK5, Biolegend, Concentration (1:200), Cat# 905501; RRID:AB_2565050; CK8, Concentration (1:100), Abcam, Cat# ab192468; CK14, Thermo Fisher, Concentration (1:200), Cat# MA5-11599; RRID:AB_10982092; NCAM1, Abcam, Concentration (1:100), Cat# ab215981; ECAD, Abcam, Concentration (1:100), Cat# ab201499; ERG, Abcam, Concentration (1:50), Cat# ab214796; CD31, Abcam, Concentration (1:100), Cat# ab218582; RRID:AB_2857973; CD3, Concentration (1:100), Abcam, Cat# ab213608; Chromogranin A, Concentration (1:100), Abcam, Cat# ab215276; AR, Cell Signalling Technology, Concentration (1:100), Cat# 8956; RRID:AB_11129223. Slides were incubated either for 2 h at room temperature or overnight at 4 °C and washed four times in 1X PBS for 5 min. Tissue sections were mounted with Slowfade Gold DAPI mounting media and imaged using a Zeiss Axioscan.Z1 microscpope. After a successful scan was obtained, the fluorophore signal was quenched via soaking slides in a quenching solution (10% 10X PBS, 0.4% 5 M NaOH, 3% $H_2O_2$) under broad-spectrum light for an hour. Slides were washed three times in 1X PBS for 5 min and mounted in Slowfade Gold DAPI mounting media and imaged with Axioscan to confirm quenching of the signal. We acquired images of each slide post-quenching to measure the autofluorescence levels at rounds 3 and 6 (Supplementary Fig. 9).

**Antibody conjugation**. Buffer exchange has been done using Amicon ultra 10 KDa spin columns for antibodies that had sodium azide as preservative. Alexa Fluor dyes were prepared by dissolving each dye in DMSO to a final concentration of 10 mM. Each antibody was mixed with 1 M NaHCO3 in 10:1 volume ratio. 0.6 μl of Alexa Fluor dye was added per 100 μg of antibody. Conjugation reaction was carried out on a rocker for 2 h at room temperature in dark. A buffer exchange using Amicon ultra 10 KDa spin columns was performed to remove the excess Alexa Fluor dye.

**Cell lines**. PC3 (ATCC CRL-1435) and LNCaP (ATCC FGC CRL-1740) human prostate epithelial cells were maintained in RPMI 1640 medium with 10% FBS at 37 °C and 5% $CO_2$ at recommended densities. Adherent cells were detached using TrypLE Express (Gibco) and were collected at mid-log phase for all experiments. After collection, cells were washed twice with ice cold 1X PBS. Cells were then filtered with a 35 μm cell strainer (Corning). Cell viability and concentration were measured with Trypan blue on the Countess II FL (Life Technologies). Cell viability was greater than 90% for all samples.

**Cell line fixation, staining and imaging**. PC3 cells and LnCAP cells were plated at 20k/well and 40k/well concentrations respectively on 96 well plates with #1.5 polymer coverslip bottoms (Cellvis). After reaching 80% confluence, cells were washed once with 1X DPBS(-) and fixed with 4% PFA diluted from a freshly opened 16% PFA (Electron Microscopy Sciences) ampoule, for 15 min at RT. Fixed cells were washed with 1X DPBS(-) once and permeabilized and blocked in a 3% BSA solution (Thermo Fisher) with 0.5% Triton-X-100 (Sigma-Aldrich). Primary antibodies for NLGN1 and NRXN1 were prepared at 1:100 dilution in 3% BSA solution and incubated with cells for 2 h at RT or overnight at 4 °C. Secondary antibodies were prepared at 1:500 dilution in 3% BSA and incubated for 1 h at RT, followed by application of 0.01 mg/mL DAPI in 1X PBS (Thermo Fisher) (5 mg/ml diluted 1:500) for 10 min at RT. After each staining step cells were washed with 1X DPBS(-) three times. Cells were imaged using a Zeiss/Yokogawa CSU-X1 spinning disk confocal setup with a 40× objective.

### sci-ATAC-seq data analysis and visualization

*Read alignment and pre-processing*. Analysis of reads was done using snapATAC (Single Nucleus Analysis Pipeline for ATAC-seq)[39]. Bases were converted to fastq format using bcl2fastq. Reads were then aligned using snaptools (v1.1) aligned-paired-end command using all of the default parameters. Reads were aligned to hg19 using the – bwa parameter, which utilizes the bwa aligner[87]. Reads were than preprocessed using the snaptools snap-pre command using the default parameters: -min-mapq=30 -min-flen=0 -max-flen=1000 -keep-chrm=TRUE -keep-single=TRUE -keep-secondary=False -max-num=1000000 -min-cov=100.

*sci-ATAC-seq binned counts matrix and peak counts matrix generation and quality control (QC)*. Two count matrices were used in this study, one being a binned matrix where each counts were generated for each 5 kb bin in the genome. This matrix was used for clustering. The binned counts were generated using the snaptools - snap-add-bmat command. For the peak matrix, the R package snapATAC command runMACS was used using the MACS2[88] parameters: -nomodel -shift 100 -ext 200 -qval 5e-2 -B –SPMR. The peak matrix was then generated and added using the createPmat function in the snapATAC. To ensure that peaks were not dominated by high input samples, peaks were called on: (1) every sample individually, (2) each cluster individually, (3) the entire combined dataset as a whole. Then all peaks were combined into one master peak matrix which was used for downstream processing. This master peak set consisted of 125,569 peaks.

*Single-cell clustering and visualization*. The matrix of binned counts was used binarized and inputted into a latent dirichlet allocation dimensionality reduction utilizing the method described by the tool cisTopic[40]. This was done using the snapATAC runLDA function using 30 topics. The Uniformed Manifold Approximation and Projection (UMAP) algorithm was then applied to the top 30 topics using the snapATAC runUMAP function. To further cluster the cells, the runCluster function was applied utilizing the Louvain method for community network detection[89].

*Differentially accessible regions analysis*. Differential accessibility was performed on the MACS2 callpeak matrix using the findDARs function in snapATAC. This function utilizes the edgeR package using the exactTest method[90]. P values were than adjusted using the Benjamini−Hochberg method. Two differentially accessible lists were generated for each comparison, one with p values < 0.05 and another with FDR < 0.05.

*GREAT analysis*. To further understand the epigenetic factors in cell groups, the top 1500 peak regions per topic were saved and fed into Genomic Regions Enrichment of Annotations Tool (GREAT v4.0.4). Further, differentially accessible regions from individual comparisons were also used as input to GREAT[46].

*HOMER analysis*. Motif enrichment was done using the homer findMotifsGenome.pl command using a bed file of interest with the parameters: -size 200 -mask using HOMER v4.11[53].

*Cicero analysis*. Cis-regulatory interactions and co-accessibility scores were plotted with the R package using the function plot_connections using Cicero 1.0.0[51].

*Silhouette analysis*. Silhouette Coefficients were calculated per cell for each sample using the R package cluster with the function silhouette. A distance matrix was used as input from the topic-cell matrix using the base R dist function.

*TCGA data analysis*. TCGA bulk ATAC-seq and RNA-seq data were downloaded and analyzed from gdc.cancer.gov filtering for PRAD datasets. Data were then normalized and analyzed for differential accessibility and differential gene expression using the edgeR package.

**Image registration and background subtraction**. Raw TIFF image files from each round of cyclic IF imaging were registered using the nuclear DAPI image from the first round as reference[91]. Background subtraction was done using a blank imaging cycle with no fluorescently tagged antibodies to remove the tissue autofluorescence signal from the images using a matrix subtraction operation[91].

**Single-cell segmentation, feature extraction and quantification**. DAPI image from the very last round of imaging was used to account for the tissue loss over cycles. Nuclei and cytoplasm segmentation were done using the QiTissue software (http://www.qi-tissue.com/) with the following parameters: Nuclei segmentation method (Advanced Morphology for Tissue), Use nuclei cycle (8), Detection Sensitivity (100%), Min/Max Diameter (28/29 pixels), Separation Force (100%), Cytoplasm segmentation method (Donut), Detection Sensitivity (100%), Max Diameter (150 pixels), Perinuclear region (3 pixels), Donut width (10 pixels), Neighbour Touch Region (8 pixels).

Average signal intensity of each marker in nucleic and cytoplasmic compartments, nucleic $x$ and $y$ coordinates, nucleus and cell size features were extracted for each segmented cell using QiTissue's "Measure Cell Features" options.

Cells with abnormal size features that did not fall within approximately the 5-to-95 percentile range were assumed to be incorrectly segmented, either multiple cells merged together or a fragment of the cell segmented and excluded from the analysis.

Three regions of interest (ROIs) were selected from each tissue section based on the H&E images reviewed by a pathologist. ROIs were selected to avoid areas of tissues with edge effects or staining gradients. Within each ROI thresholds for NRXN1, NLGN1 and different cell type markers were determined. Using the thresholds for cell type markers, we identified luminal (CK8), basal (CK5, CK14) and neuroendocrine (Chromogranin A) epithelial cells, immune cells (CD3), endothelial cells (CD31) and neuronal cells (N-CAM). Based on the cell type and marker (NRXN1 and NLGN1) thresholds, we identified cells that express only NRXN1, only NLGN1, both NRXN1 and NLGN1 or neither for each cell type. The quantities of each of these categories for each cell type was plotted using the "ggplot2" library of R, version 3.4.3.

**Survival analysis**. We used the Cancer Transcription Factor Targets pipeline within the Cistrome Cancer ("the set of cis-acting targets of a trans-acting factor on a genome-wide scale, also known as the in vivo genome-wide location of [transcription factor binding-sites] or histone modifications")[54]. Survival Analysis was plotted using the TCGA PRAD data set, selecting for transcription factors FOXA1, HOXB13 and CDX2.

**Reporting summary**. Further information on research design is available in the Nature Research Reporting Summary linked to this article.

## Data availability
The raw single-cell ATAC-sequencing files and processed data files generated in this study are available under the super series GSE accession number: GSE171559. The TCGA bulk ATAC-seq data used in this study are available here: https://gdc.cancer.gov/about-data/publications/ATACseq-AWG. The TCGA bulk RNA-seq publicly available data used in this study are available in the gdc database under accession code: prad-2015. hg19 reference genome used in the study is available here: https://www.ncbi.nlm.nih.gov/assembly/GCF_000001405.39. Source data are provided with this paper. The remaining data are available within the Article, Supplementary Information or Source Data file. Source data are provided with this paper.

## Code availibility
Code used for the analysis of sci-ATAC-seq data in this study is available on Github (https://github.com/AlexChitsazan/ProstateTumorATACCode). Code used for cyclic IF analysis in this study is available on GitHub (https://github.com/zeynepsayar/Neuronal_cyclic_R). Both codes are also deposited to Zenodo: https://zenodo.org/record/5644071#.YYLw7NbMI8M and the corresponding DOI is as follows: https://doi.org/10.5281/zenodo.5635457. Source data are provided with this paper.

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

## Acknowledgements
We are grateful to all patients who participated to the OHSU study (IRB #18321) that provided tissue from prostatectomy whole-mounts. R.P.K. was funded by the Collins Medical Trust for this study. We are also grateful to the Biolibrary and the Histopathology Shared Resource (HSR) at OHSU for providing patient samples, specifically Aletha Letsch and Cheyenne Martin. The Biolibrary and HSR was supported by NIH grants P30 CA069533 and P30 CA069533 13S5 through the Knight Cancer Institute. We thank Erin Watson for collecting patient information. We thank the Spellman lab and the prostate cancer working group in CEDAR for engaging discussions. We are grateful for the collaborative environment in the Knight Cancer Institute and productive interactions with the Grey, the Chang and the Adey labs. We thank Kemal Sonmez for his input on single-cell data analysis. We thank Crystal Shaw from the Advanced Light Microscopy Core and Dorian LaTocha from the Flow Cytometry Core at OHSU for their technical help. We thank Shelley Barton, Hisham Mohammed, Joshua Saldivar, Stefanie Linch and Lindsey Minter for providing feedback on the manuscript. This project was supported by funding (CEDAR3410918) from the Cancer Early Detection Advanced Research Centre at Oregon health & Science University, Knight Cancer Institute (S.E.E.).

## Author contributions
S.E.E., A.A. and P.T.S. conceived the study. S.E.E. performed all experiments with help from Z.S. and A.F. sci-ATAC-seq data analysis was performed by A.C. and A.A., while A.C. and S.E.E. performed the analysis of TCGA data. Z.S. and S.E.E. performed the analysis of cyclic IF data. G.T. and R.P.K. recruited patients and collected whole-mount patient samples. G.V.T. reviewed the pathology. S.E.E. processed all patient samples. S.E.E., Z.S., A.C., G.T., A.A. and P.T.S. performed the biological analysis and interpretation. S.E.E. wrote the manuscript with input from all authors.

## Competing interests
The authors declare no competing interests.
