## [Peer Review File · Nature Communications]

Epigenetic loss of heterogeneity from low to high grade localized prostate tumoursReviewers' Comments:

Reviewer #1:

Remarks to the Author:

The stated goal of the manuscript entitled 'Single-cell analysis of localized low- and high-grade prostate cancers' by Eksi et al was to 'identify molecular and cellular markers associated with prostate tumors that have a primary Gleason pattern 3 (low-grade) and 4 (high-grade) to explore the heterogeneity of the localized disease'. The paper is fairly written, but the figures are poorly constructed and very difficult to analyze. The main negative drivers of the impact of this study are the low patient numbers and the lack of deeper analysis of the loss of heterogeneity from Gleason 3 to Gleason 4. The paper and figures are constructed as if the data should just be downloaded by the reader to figure out for themselves. A major rethinking of the presentation and interpretation of the data is needed.

1. The type font in nearly every figure is too small to read. It looks as if each figure panel was simply shrunk to fit into a dedicated space without concern for whether it would be readable. Saving your R files as .eps and investing in Adobe Illustrator to adjust fonts is a must.
2. The annotations of cell type in supp figure 1 is entirely inadequate. Improved software is readily available for clustering and annotation of cell type/state from scATAC-seq data using existing references (PMID 33637727). The annotation of 'topics' with GREAT is simply confusing. Even so, an attempt should have been made to apply the deduced cell type names to each cluster instead of leaving them as 'cluster 7, 8, 9, etc.'
3. Gleason 3 tumors are stated to mostly share open chromatin profile because they form a 'single' cluster in UMAP space, but the subclusters within that main clusters look very patient-specific. This is actually the most important point of the study: the heterogeneity of Gleason 3 is reduced by Gleason 4.
4. An interesting topic to pursue is whether you can identify open chromatin profiles in a subset of cells within the Gleason 3 patients that could predict progression. Surely some of the cells in Gleason 3 have for neuronal adhesion molecules. There is a story here that needs to be developed further.

Reviewer #2:

Remarks to the Author:

In the manuscript "Single-cell analysis of localized low- and high-grade prostate cancers" Sebnem Ece Eksi et al. studied chromatin accessibility in prostate cancer in 14,424 single cells from 18 patients using single cell ATAC-seq to overcome tumor cell heterogeneity. The authors show differences in chromatin between low- and high-grade prostate tumors. Differential accessible sites were found at neuronal gene loci NRXN1 and NLGN1 which were broadly expressed in stromal and epithelial cells in prostate tumors.

This is an interesting study, but the potential immense additional value of this study over TCGA bulk ATAC-seq analysis is at the moment severely limited due to the low number of cells for individual samples which are a challenge for clustering and detection of differences between groups. Please see major points below:

Major Points:

- 1) I am wondering why for most samples there is only a few hundred nuclei that pass quality control and large fraction of nuclei ~26% is contributed by one sample (15)? With these low numbers of cells for most samples the cellular heterogeneity in tumors from different patients is hard to assess and it is not clear how to interpret the findings. Since per nucleus only ~ 1287.5 fragments the power to detect peaks and perform differential analysis between groups is limited (e.g. only a few hundred thousand total read for a whole tumor sample). This makes interpretation of differential sites very difficult, e.g. it is surprising that there are no sites with lower accessibility in G4 stages. It was also not clear what is the difference between differential analysis shown in Fig 3 and Supp Fig. 2A/B. Are these for direct comparison between outer clusters with the inner cluster? Why are there only differential sites for one

comparison? From the methods it is not exactly clear how the differential elements were detected and what covariates such as patient, age besides Gleason stage were used. To evaluate the differential sites at the NLGN1/NRXN1/CDH9 loci, please show genome browser tracks and sample resolved heatmaps. Panel 3E could be omitted or modified, since as the text states it does not highlight the differential distal sites but promoters. The potential issue with the differential analysis presented here becomes apparent in the comparison to bulk ATAC-seq: According to the genomic coordinates the sites at the NRXN1 and CDH9 loci that were differential in bulk comparison are different than the ones detected from the single cell analysis. Were there no sites detected from NLGN1 locus in bulk comparison? Overall, this seems to indicate that the results identified in this study cannot be supported in the larger cohort of 26 patients. How do the authors reconcile these discrepancies? How many total differential sites were identified from bulk ATAC-seq and how many of these were found in the single cell ATAC-seq data?

Was NRXN1 higher expressed in high-grade compared to low grade tumors (which would be expected based on the higher accessibility detected)? The authors state that NLGN1 was not detected in RNA-seq in TCGA; what about CDH9? And how does it relate to the cyclic IF data?

2) Overall, it is not clear how tumor cell clusters were identified, e.g. how were normal epithelial cells distinguished from cancerous epithelial cells? The authors state in line 212-214 that Gleason 3 tumors formed without patient-specific clusters and point to Supp Figure 4, but Supp Figure 4 shows 13 clusters (panel B) and individual samples are located to distinct places on the UMAP. Would this not indicate patient-to-patient differences? ERG motif also seems to be enriched mostly in a subset of cells which shows the heterogeneity within this large cluster. Supp Fig 1 B indicates that indeed most of the clusters are dominated by nuclei from one patient.

General Response

We thank the reviewers for their valuable comments and suggestions. Major updates in the new version of the manuscript include:

- new analysis of loss of heterogeneity from Gleason pattern 3 to 4
- detailed cluster annotation
- additional differential chromatin accessibility analysis performed in the absence of Sample_15 for further quality assessment of the sci-ATAC-seq data
- a detailed explanation and new figure panels about the discrepancies observed with the TCGA's bulk data

Minor revisions include:

- re-creation of all figures via Adobe Illustrator for better data presentation as suggested by Reviewer 1
- addition of a new bar graph showing cell counts for each cluster in Gleason pattern 3 cisTOPIC analysis
- new supplementary figure showing genome browser tracks for differentially accessible sites
- a new figure panel showing sample resolved heatmap
- additional figure titles and legends to guide readers through TCGA's bulk ATAC-seq analysis
- change of the immunofluorescent image in Figure 5: even though this change did not change our conclusion, we were able to acquire higher quality images of the tissues
- reorganization of specific figure panels and addition of a new subsection to the Results section for improved manuscript flow

We believe the manuscript improved significantly based on the feedback we received from both of the reviewers and we are thankful for their thoughtful comments.

We include a version of the manuscript with changes tracked and a second version with all changes accepted. Below we include a point-by-point response.

REVIEWER COMMENTS

Reviewer #1, expert in prostate cancer genomics and subtypes (Remarks to the Author):

The stated goal of the manuscript entitled 'Single-cell analysis of localized low- and high-grade prostate cancers' by Eksi et al was to 'identify molecular and cellular markers associated with prostate tumors that have a primary Gleason pattern 3 (low-grade) and 4 (high-grade) to explore the heterogeneity of the localized disease'. The paper is fairly written, but the figures are poorly constructed and very difficult to analyze.

We thank the reviewer for their valuable suggestions and helpful feedback on the manuscript and the figures. We revised **all** figures in the paper based on their feedback, using Adobe Illustrator. We changed the text labels and included more description of the data throughout the paper.

The main negative drivers of the impact of this study are the low patient numbers...

Unfortunately, single-cell ATAC-seq requires flash-frozen tumor samples and we do not expect to have access to more radical prostatectomy tumor samples for single-cell ATAC-seq analysis at the moment. We completely agree with the reviewer that sequencing more cells from a bigger cohort of patients would give us an opportunity to catalogue the molecular landscape of localized prostate tumors in further detail. We aim to find a new patient cohort and expand our study to more patients in the next years. However, we think our study from 18 patients specifically focused on two biological groups, primary Gleason pattern 3 and 4 prostate tumors, reveal unique features about the biology of the localized disease, specifically the identification of two unique neuronal adhesion molecules NRXN1 and NLGN1, which was validated using immunofluorescent microscopy and the transcription factor binding motifs enriched in high-grade localized prostate tumors, FOXA1, HOXB13 and CDX2, which was validated using the TCGA data set. As compared to other single-cell ATAC-seq studies that may have increased sequencing depth as a result of the tissue-of-origin, our study provides an in-depth validation of identified targets at the protein expression level using an orthogonal approach, cyclic immunofluorescent microscopy.

...and the lack of deeper analysis of the loss of heterogeneity from Gleason 3 to Gleason 4. The paper and figures are constructed as if the data should just be downloaded by the reader to figure out for themselves. A major rethinking of the presentation and interpretation of the data is needed.

We thank the reviewer for pointing out that our loss of heterogeneity observation could benefit from further analysis. We further investigated the intra-sample and inter-sample heterogeneity of Gleason pattern 3 and 4 tumors based on their feedback and obtained new data that strongly supports our hypothesis in the new version of our manuscript. We performed silhouette analysis using topic modeling to measure the distances between each Gleason pattern 3 and 4 sample. Our results show how each cell in a Gleason pattern 4 cluster is separated from cells in Gleason pattern 3 cluster, providing a measure of the decreased heterogeneity of Gleason pattern 4 tumors (Figure 3F). We also aggregated all single cells from Gleason pattern 3 vs. 4 tumors and observed a higher separation of the Gleason pattern 4 tumors as compared to Gleason pattern 3. We added this new data as a Figure in the revised manuscript (Figure 3G-H). We also added a new subsection to our results section to highlight this observation and new analysis:

“Loss of heterogeneity from Gleason pattern 3 to 4

Our cluster analysis show Gleason pattern 4 tumours as the outlier clusters on the UMAP whereas cells from primary Gleason pattern 3 tumours are mixed with each other in one cluster, regardless of patient ID. To further investigate the loss of heterogeneity from Gleason pattern 3 to 4 prostate tumours, we performed Silhouette analysis to measure the distances between aggregated cells from Gleason pattern 3 and 4 tumours. Our results show cells from Gleason pattern 4 tumours to have decreased heterogeneity (Figure 3F). To measure the level of heterogeneity within each primary Gleason pattern 3 and 4 sample, we also examined the Silhouette scores separately and found cells from Gleason pattern 3 tumours to consistently have higher heterogeneity as compared to Gleason pattern 4 (Figure 3F). To determine how this loss of heterogeneity is observed at the level of trans-regulators, we decided to examine the transcription factor (TF) binding motif analysis in Gleason pattern 3 and 4 prostate tumours.”

F. Silhouette analysis using topic modelling for single-cells aggregated from each Gleason score 3+3 (red) and 4+4 (blue) prostate resolved by patient samples and all Gleason pattern 3 (red) and 4 (blue) prostate tumours aggregated.

Along with our TF motif analysis that shows each Gleason pattern 4 tumor converging on a similar trans-regulatory landscape that are distinct from Gleason pattern 3 tumors and our cluster analysis that shows Gleason pattern 4 tumors as the outlier clusters on the UMAP, our paper now provides three lines of evidence for the loss of heterogeneity from Gleason pattern 3 to 4 prostate tumors.

1. The type font in nearly every figure is too small to read. It looks as if each figure panel was simply shrunk to fit into a dedicated space without concern for whether it would be readable. Saving your R files as .eps and investing in Adobe Illustrator to adjust fonts is a must.

The font in all figure legends is formatted and all figure panels are re-arranged using Adobe Illustrator.

2. The annotations of cell type in supp figure 1 is entirely inadequate. Improved software is readily available for clustering and annotation of cell type/state from scATAC-seq data using existing references (PMID 33637727). The annotation of 'topics' with GREAT is simply confusing. Even so, an attempt should have been made to apply the deduced cell type names

to each cluster instead of leaving them as 'cluster 7, 8, 9, etc.

We thank the reviewer for this important feedback. We provided more information about the clusters in Figure 2E and Supplementary Figures 2 and 3 in the new version of the manuscript.

We are aware of two ways of annotating single-cell ATAC-seq clusters and we used both of them in our analysis: 1) gene-based annotation tool of snapATAC, which infers gene expression based on the chromatin accessibility level on the gene body 2) GREAT analysis of the Topics identified.

To make sure there are not any additional cluster annotation tools we are missing in our analysis we talked with Dr. Bing Ren, the corresponding author of the snapATAC publication. We are using the gene-annotation tool he mentioned below for cluster annotations. Please see our correspondence with Dr. Ren about the cluster annotation feature of snapATAC.

“From: "Ren, Bing" <biren@health.ucsd.edu>
Date: Wednesday, June 23, 2021 at 7:51 AM
To: Ece Eksi <eksi@ohsu.edu>, Bing Ren <biren@ucsd.edu>
Subject: [EXTERNAL] Re: snapATAC cluster analysis

Dear Sebnem,

The gene-based annotation is in the Github (https://github.com/r3fang/SnapATAC/blob/master/examples/10X_brain_5k/README.md#gene_tsne). The annotation function is to infer the gene expression for marker genes based on the chromatin accessibility level at the gene body. Based on the chromatin accessibility score at the marker genes, one can annotate clusters.

Best,

Bing Ren, Ph.D.
Professor, Cellular and Molecular Medicine
Director, Center for Epigenomics
UC San Diego

Member
Ludwig Institute for Cancer Research

It is important to note here that annotating all clusters with snapATAC using cancer samples is very difficult as compared to cell type annotations done with healthy brain samples that consists of very well differentiated cell types. In our sci-ATAC-seq experiments, we used Gleason grade 3 and 4 prostate cancers, which by definition consists of less differentiated glandular epithelial cells. As a result, we observe that all cells in our data set, once the two immune cell clusters and the stromal cluster are removed, are epithelial prostate cancer cells. Chromogranin A cells that normally compromise approximately 1% of the epithelial cell population did not form a separate cluster (Supplementary Figure 2A). Similarly, basal epithelial cells were observed to be scattered across the luminal epithelial cell clusters (Supplementary Figure 2A). To bolster our cell-type identification, we also examined the accessibility for key transcription factors, including

ERG, HOXD13, ID4, L3MBTL4, ZNF154, ZNF655, DLX1-6. We identified C-MYC, ERG and FLI1 enriched clusters in our data set (Supplementary Figure 2).

We revised Figure 2E to show a more detailed annotation of clusters. We generated a new Supplementary Figure (Supplementary Figure 2) to show the gene-based annotations of prostate cell type markers used in cluster annotation. Panel A shows the main epithelial cell type markers luminal (KRT8), basal (KRT5 and KRT14) and neuroendocrine (Chromogranin A). Panel B shows other common prostate epithelial cell markers KLK3 (PSA), AR, TMPRSS2 and NKX3.1. Panel C shows the markers commonly used to identify prostate cancer cells AMACR, EPCAM, CDH1 and PTEN as well as the proliferation marker Ki67. We observed high accessibility for C-MYC in Gleason pattern 4 clusters, which is now included in our gene-based annotation map (Supplementary Figure 2C). We used GO terms identified with GREAT to annotate the heterogeneous cluster of cells, since we did not observe any specific genes that would define all cells in these clusters. However, we observed that, despite the heterogeneity of cells in the Gleason pattern 3 cluster, all clusters are all enriched for GO terms associated with an inflammatory response: wound healing (Clusters 8, 10), leukocyte activation (Clusters 4, 11), viral response (Clusters 6, 9, 16) and IL6 signaling (Clusters 1, 5).

We believe the additional cluster annotations we provided in the new version will help readers navigate the presented data more easily.

We also added a new subsection that describes cluster annotations in the manuscript:

“Epithelial prostate cancer cells carry markers of inflammatory response

We used GREAT analysis in addition to the gene-based annotation tools of snapATAC to annotate the rest of the clusters identified through the topic analysis. We inferred the gene expression of marker genes for prostate tissue KRT8 (luminal epithelial), KRT5 (basal epithelial), KRT14 (basal epithelial), Chromogranin A (neuroendocrine), KLK3 (luminal epithelial), AR, TMPRSS2 and NLX3.1 (Supplementary Figure 2A-B). We also examined markers for prostate cancer such as AMACR, EPCAM and C-MYC as well as molecular markers CDH1 and PTEN (Supplementary Figure 2C). Our results show that the majority of our cells come from luminal epithelial cells as expected, with high accessibility to cancer markers AMACR and EPCAM (Supplementary Figure 2C). Basal and neuroendocrine cells, which normally constitute a small percentage of the epithelial cell population, do not form independent clusters on the UMAP (Supplementary Figure 2A-B). GREAT analysis shows an enrichment for regions associated with early inflammatory response in cells that formed the main cluster island on the UMAP. Topics 9, 15, 19 and 21 define the cells in the main cluster, which include gene terms related to IL-6 signaling, wound healing, viral response and leukocyte activation, respectively (Figure 2C). We observed high accessibility for C-MYC for the three outer clusters (Supplementary Figure 2C). One of these outer clusters, also shows high accessibility for ERG and the other for FLI1 (Supplementary Figure 2D). We decided to further delineate the biological differences between the main cluster and the three distant clusters of cells on the UMAP (Figure 2E).”

Figure 2E. Epithelial and stromal cell types are identified in localized prostate tumors. Lymphoid, myeloid and fibroblasts (gray tones) are removed from downstream analyses. Outer clusters show high-MYC accessibility (blue tones). Middle cluster shows higher accessibility to genes associated with inflammatory response (red tones).

Finally, we prepared a new supplementary Figure that shows the detailed gene-annotations for key cell-type markers in prostate cancer:

“Supplementary Figure 2: Annotation of clusters using inferred gene expression. A. Epithelial cell type markers luminal (KRT8), basal (KRT5 and KRT14) and neuroendocrine (Chromogranin A). **B.** Common prostate epithelial cell markers KLK3 (PSA), AR, TMPRSS2 and NKX3.1. **C.** Prostate cancer cell markers AMACR, EPCAM, CDH1, C-MYC and PTEN as well as the proliferation marker Ki67. **D.** 16 Clusters that were identified around 30 Topics exhibit hierarchical clustering as shown by a cluster dendrogram. Clusters 12 and 14 consisted of immune cells, which formed a separate cluster in the dendrogram. Cluster 7 consisted of stromal cells associated with the prostate cancer epithelial cells. Clusters in the main island are enriched for GO terms associated an inflammatory response: wound healing (Clusters 8, 10), leukocyte activation (Cluster 11), viral response (Clusters 4, 6, 9, 16) and IL6 signalling (Clusters 1, 5). Outer clusters show high accessibility for C-MYC (Clusters 2, 3, 13, 15). Cluster 2 show accessibility for FLI1 and Cluster 3 for ERG.”

3. Gleason 3 tumors are stated to mostly share open chromatin profile because they form a 'single' cluster in UMAP space, but the subclusters within that main clusters look very patient-specific. This is actually the most important point of the study: the heterogeneity of Gleason 3 is reduced by Gleason 4. An interesting topic to pursue is **whether you can identify open chromatin profiles in a subset of cells within the Gleason 3 patients that could predict progression**. Surely some of the cells in Gleason 3 have for neuronal adhesion molecules. There is a story here that needs to be developed further.

We thank the reviewer for this feedback. We definitely observe some cells from Gleason pattern 3 tumors with higher NRXN1 and NLGN1 accessibility, which may mark these cells as more aggressive (Supplementary Figure 4A), specifically cells in Clusters 1, 5 and 9. When we test NRXN1 and NLGN1 expression at the protein level, Gleason pattern 4 tumors seem to have higher expression of neuronal adhesion molecules as compared to Gleason pattern 3 (Figure 5A-B). This suggests that some cells in Gleason pattern 3 tumors acquire chromatin accessibility changes near genes encoding neuronal adhesion molecules that may result in changes in protein expression in later stages of the disease. Another interesting observation here is that some stromal cells (immune, neuronal and endothelial) also express NRXN1 and NLGN1 (Figure 5C), suggesting neuronal adhesion molecules we identified in this paper may be mediating a crosstalk between cancer cells and stromal cells, initiating an aggressive transformation.

Identifying how crosstalk between prostate cancer cells and stromal cells may be mediated by neuronal adhesion molecules NRXN1 and NLGN1 and how this interaction may result in further chromatin profile changes in Gleason pattern 3 and 4 tumors is a long-term question we would like to pursue in our lab. However, we believe this question is beyond the scope of this paper, which is primarily focused on the single-cell chromatin accessibility profiles of localized prostate tumors.

Reviewer #2, single cell ATAC-seq (Remarks to the Author):

In the manuscript “Single-cell analysis of localized low- and high-grade prostate cancers” Sebnem Ece Eksi et al. studied chromatin accessibility in prostate cancer in 14,424 single cells from 18 patients using single cell ATAC-seq to overcome tumor cell heterogeneity. The authors show differences in chromatin between low- and high-grade prostate tumors. Differential accessible sites were found at neuronal gene loci NRXN1 and NLGN1 which were broadly expressed in stromal and epithelial cells in prostate tumors.

This is an interesting study, but the potential immense additional value of this study over TCGA bulk ATAC-seq analysis is at the moment severely limited due to the low number of cells for individual samples which are a challenge for clustering and detection of differences between groups.

We thank the reviewer for their thoughtful comments. We believe incorporating their feedback greatly improved the quality of our manuscript. Please see our detailed responses below.

Please see major points below:

Major Points:

1) I am wondering why for most samples there is only a few hundred nuclei that pass quality control and large fraction of nuclei **~26% is contributed by one sample (15)**?

We thank the reviewer for this valuable feedback. To test whether our results were primarily driven by one sample (Sample_15), we added two additional analyses to the revised manuscript (Supplementary Figures 4B and 5). First, we omitted Sample_15 from our data set and performed differential accessibility analysis in the absence of this sample. We observed NRXN1, NLGN1 and CDH9 as the top genes that are significantly more accessible in Gleason pattern 4 vs. 3 tumors (Supplementary Figure 4C). We think this additional analysis proves the robustness of our differential accessibility results between Gleason pattern 3 and 4 tumors. We then looked at the sample resolved genome browser tracks at NRXN1, NLGN1 and CDH9 loci (Supplementary Figure 5), which clearly demonstrates the higher number of peaks observed in

Gleason pattern 4 tumors. Finally, when Sample_15, Sample_3 and Sample_4 are all omitted from the cisTOPIC analysis, we observe similar distributions of cells across clusters. To clarify this point, we included a new figure that shows the cell numbers per cluster in our analysis with Gleason pattern 3 tumors (Supplementary Figure 6C). We think these three lines of evidence show the biological results we obtain from the analysis are not skewed as a result of the higher number of nuclei that comes from Sample_15. We think these additions to the manuscript improved the quality of the work so we thank the reviewer for pointing this out.

With these low numbers of cells for most samples the cellular heterogeneity in tumors from different patients is hard to assess and it is not clear how to interpret the findings. Since per nucleus only ~ 1287.5 fragments the power to detect peaks and perform differential analysis between groups is limited (e.g. only a few hundred thousand total read for a whole tumor sample).

Performing single-cell ATAC-seq using primary prostate tumors results in smaller number of cells with unique fragment reads. Even though 1000 fragments per nucleus seems small, the calling of peaks over all samples is robust and the per cell assessment is then, while incomplete, robust for major features. Our orthogonal experimental approach, cyclic immunofluorescent microscopy, shows that the differentially accessible genomic regions identified in the study drive expression of neuronal adhesion gene NRXN1 and NLGN1 in prostate cancer tissue-sections.

This makes interpretation of differential sites very difficult, e.g. it is surprising that there are no sites with lower accessibility in G4 stages.

We thank the reviewer for pointing this out. Gleason pattern 4 tumors are less differentiated as compared to Gleason pattern 3 tumors and it is entirely possible that they have more regions that are open without closing any regions. This is a model that fits our understanding of tumor evolution in the lab.

It was also not clear what is the difference between differential analysis shown in Fig 3 and Supp Fig. 2A/B. Are these for direct comparison between outer clusters with the inner cluster? Why are there only differential sites for one comparison?

We thank the reviewer for this comment. We agree that the previous version of the manuscript did not clearly explain the differences between Fig 3 and Supp Fig 2A/B. We believe the changes we made in the revised version clarified our analysis results and how we present them in the paper. Fig 3 shows differentially accessible regions with a less stringent statistical value, i.e., the top 1033 sites in Gleason pattern 4 tumors. In Supp Fig 2A (previous version) we adopted a more stringent statistical approach, showing only the top 23 sites in Gleason pattern 4 tumors, which shows a similar enrichment for neuronal adhesion molecules. We realize that including both statistical approaches is not necessary, especially because they show very similar results. Therefore, we eliminated Supp 2A to prevent further confusion in the revised version of our manuscript. Supp 2B (previous version; Supp Fig 4C in the revised version) shows accessibility sites significantly enriched in Gleason pattern 3 tumors as opposed to Gleason pattern 4, the top 279 regions in Gleason pattern 3 prostate tumors. Together these data show direct comparisons between outer vs. inner and inner vs. outer clusters.

From the methods it is not exactly clear how the differential elements were detected...

We thank the reviewer for pointing this out. We agree that further explanation was needed in the Methods section to clarify the analysis. We changed the methods section as follows to provide more information about the detection of differential elements:

“To ensure that peaks were not dominated by high input samples, peaks were called on: 1) every sample individually, 2) each cluster individually, 3) the entire combined dataset as a whole. Then all peaks were combined into one master peak matrix which was used for downstream processing. This master peak set consisted of 125,569 peaks.”

... and what covariates such as patient, age besides Gleason stage were used.

We did not use any other variables beside Gleason grade to direct our analysis. However, we calculated CAPRA-S and nomogram scores for all patients in our cohort to predict potential patient outcomes and to make comparisons with the TCGA's cohort in terms of clinical variability (Supplementary Table 1). We made the following changes in the methods section to clarify this point

“...These factors were used to determine the specific clinical features of our patient cohort in comparison to the TCGA's cohort.”

To evaluate the differential sites at the NLGN1/NRXN1/CDH9 loci, please show genome browser tracks and sample resolved heatmaps.

We thank the reviewer for this suggestion and now include a new supplementary figure. Supplementary Figure 5 shows the genome browser tracks of each sample at the NRXN1, NLGN1 and CDH9 loci, which clearly shows the differential sites between Gleason pattern 3 and 4 tumor samples. We also added the sample resolved heatmap as a new panel to Supplementary Figure 4A.

Supplementary Figure 4A. Heatmap showing Topic enrichment in each sample.

A

B

C

Supplementary Figure 5: Genome browser tracks showing peaks at the NRXN1 (A), NLGN1 (B) and CDH9 (C) loci. Samples with primary Gleason pattern 4 tumours are highlighted in blue.

Panel 3E could be omitted or modified, since as the text states it does not highlight the differential distal sites but promoters.

We agree with the reviewer. We included panel 3E as a supplementary figure (Supplementary Figure 4A) in the new version of our manuscript.

The potential issue with the differential analysis presented here becomes apparent in the comparison to bulk ATAC-seq: According to the genomic coordinates the sites at the NRXN1 and CDH9 loci that were differential in bulk comparison are different than the ones detected from the single cell analysis. Were there no sites detected from NLGN1 locus in bulk comparison? Overall, this seems to indicate that the results identified in this study cannot be supported in the larger cohort of 26 patients. How do the authors reconcile these discrepancies?

We thank the reviewer for their crucial feedback. In the previous version of the manuscript, we included details about the clinical grades of the patients in our cohort vs. the TCGA cohort in the Supplementary Excel Files 1 and 2, without any clear explanation in the main text. The previous version of the figures shows that TCGA data analysis did not have any clear labels with Gleason scores and clinical grades of the patients. In the revised manuscript, we included critical details about how the clinical grades of the prostate tumors in our cohort vs. the TCGA study are different from each other, providing an explanation of the differences we expect to see with the bulk TCGA data set. We added clear legends to Supplementary Figure 8 that show the specific Gleason scores of the patients used in the analysis. We highlighted the patient samples used in the differential analysis. We believe these changes provided a more crystal-clear understanding of the TCGA bulk ATAC-seq analyses provided in the manuscript.

In summary, our cohort of patients mainly consists of low- and high-grade patients with Gleason score 3+3 and 4+4 tumors: 12 patients with pT2 tumors and 5 patients with pT3 tumors (three pT3a and two pT3b). In contrast, the TCGA's cohort consist of intermediate- and high-grade patients with Gleason score 3+4, 4+3, 4+4, 4+5 and 5+4 prostate tumors: 19 patients with pT3 tumors and 6 pT2 tumors (one T2b and five pT2c). When we performed differential analysis using the TCGA cohort, we compared Gleason 4+3, 4+4 and 4+5 (high-grade) to Gleason score 3+4 patients (intermediate-grade) because TCGA did not include any ATAC-seq analyses using Gleason score 3+3 tumors. In comparison, the differential analysis with our patient cohort was done comparing Gleason 4+4 (high- and intermediate-grade) to Gleason 3+3 (low-grade) tumors. As a result of these fundamental clinical differences between the two cohorts, we do not expect to see the exact same chromatin sites being enriched in differential analyses. The analysis of the TCGA's data set should therefore be evaluated independently as a comparison between high-grade vs. intermediate-grade prostate tumors, which also suggested a similar significant trend in accessibility near genes encoding neuronal adhesion molecules.

To clarify this point we made the following changes in the manuscript:

“TCGA PRAD data set consists of 26 patients with intermediate- and high-grade prostate tumours (Supplementary Figure 8, Supplementary File 2). Only six patients have Gleason score 4+4 tumours and none of the patients have Gleason score 3+3 tumours. Despite these major clinical differences in the patient data sets, we observed a significant increase in the accessibility of NRXN1 and CDH9 chromatin sites in high-grade tumours as compared to intermediate-grade tumours (Supplementary Figure 8). Next, we analysed the bulk RNA-seq

profiles of 497 patients from the TCGA cohort and detected NRXN1 expression in the majority of the prostate tumours (Supplementary Figure 8). In contrast, we did not detect NLGN1 transcripts in the TCGA bulk RNA-seq data set (Supplementary Figure 8).”

Please also see the new Supplementary Figure 8 in the revised manuscript:

“Supplementary Figure 8: NRXN1 and CDH9 are significantly accessible in tumours with Gleason score 4+4 and 4+5 as compared to Gleason score 3+4 in the TCGA bulk ATAC-seq data set. A. Distribution of Gleason scores across patients are shown in the TCGA bulk ATAC-seq data set. x-axis shows the number of patients in the cohort, y-axis indicates the Gleason score. Patient samples with primary Gleason grades 3 and 4 are highlighted in blue.”

Additional clinical considerations about the TCGA’s cohort are: 1) PSA level not available for one patient, 2) percent biopsy cores positive for cancer is not provided, 3) clinical T grade is not reported for some of the patients. As a result of these missing data points, we were not able to calculate the CAPRA-S score for patients in the TCGA’s cohort, which partially limits our clinical evaluations. To circumvent this problem, we calculated the CAPRA score (different than CAPRA-S) to provide a comparison between the two patient cohorts. All of this detailed clinical information is highlighted in the new version of Supplementary Data 2.

Despite these differences, we detect changes in chromatin accessibility near NRXN1 and CDH9 genes in the TCGA cohort when Gleason score 4+4 and 4+5 patients were compared with Gleason score **3+4 patients**, suggesting a potential shift in the accessibility of neuronal adhesion molecules in high-grade prostate cancers relative to **intermediate-grade**. We observe that 4 out of 4 sites detected as significantly different when **Gleason score 4+5** tumors are compared to **Gleason score 3+4** match with the differential peaks identified in our data set (Supplementary Figure 3B). We also observe that 2 out of 4 sites detected as significantly different when **Gleason score 4+4** tumors are compared to **Gleason score 3+4** match with the differential peaks identified in our data set (Supplementary Figure 3B).

To eliminate differences introduced by the differences in clinical grades between the two cohorts, we decided to look at all peaks called at the NRXN1, NLGN1 and CDH9 loci in both TCGA’s bulk ATAC-seq and our sci-ATAC-seq data sets. Our results show that 18 out of 23 peaks in the TCGA’s data set match with the sci-ATAC-seq results at the NRXN1 locus. 3 out of 7 peaks in the TCGA’s data set match with the sci-ATAC-seq results at the CDH9 locus. We did

not observe any matches at the NLGN1 locus between the two cohorts. We added this new analysis as Supplementary Figure 8H.

H

Gene	Sci-ATAC-seq	TCGA bulk ATAC-seq	Bp difference	Gene	Sci-ATAC-seq	TCGA bulk ATAC-seq	Bp difference
NRXN1	Peak_59581 (-732656)	PRAD_13077 (-732627)	29	NRXN1	Peak_59536 (+681698)	PRAD_13062 (+681930)	232
NRXN1	Peak_59566 (-126050)	PRAD_13075 (-126002)	48	NRXN1	Peak_59535 (+695379)	PRAD_13061 (+695594)	215
NRXN1	Peak_59565 (-47803)	PRAD_13074 (-47727)	76	NRXN1	Peak_59531 (+888100)	PRAD_13060 (+887952)	148
NRXN1	Peak_59553 (+313937)	PRAD_13071 (+314042)	105	NRXN1	Peak_59530 (+898213)	PRAD_13059 (+898451)	238
NRXN1	Peak_59551 (+381839)	PRAD_13070 (+382023)	184	NRXN1	Peak_59529 (+905336)	PRAD_13058 (+905423)	87
NRXN1	Peak_59550 (+388931)	PRAD_13069 (+389132)	201	NRXN1	Peak_59528 (+910520)	PRAD_13057 (+910520)	0
NRXN1	Peak_59541 (+539682)	PRAD_13067 (+539723)	41	NRXN1	Peak_59526 (+960054)	PRAD_13056 (+960137)	83
NRXN1	Peak_59540 (+553909)	PRAD_13066 (+554264)	355	NRXN1	Peak_59524 (+989586)	PRAD_13055 (+989463)	123
NRXN1	Peak_59539 (+615053)	PRAD_13064 (+615206)	153	CDH9	Peak_90760 (-349238)	PRAD_31624 (-349114)	124
NRXN1	Peak_59538 (+624088)	PRAD_13063 (+623999)	89	CDH9	Peak_90749 (+389446)	PRAD_31621 (+389332)	114
				CDH9	Peak_90739 (+792862)	PRAD_31619 (+793047)	185

Supplementary Figure 8H. The list of peaks at NRXN1 and CDH9 loci from the TCGA bulk ATAC-seq data set that match with the peaks in our data set (sci-ATAC-seq). The third column shows the base pair difference between each peak.

It is important to note here that some of the discrepant changes in chromatin accessibility sites near neuronal adhesion molecules, may eventually converge on changes in trans-regulators and protein expression. Even though the number of patients in both cohorts are not very large, collecting several lines of evidence regarding transcriptional regulatory networks and protein expression is key to understanding the downstream consequences of chromatin accessibility profile shifts. Together, our downstream investigations from the same cohort of patients in this study provide strong support for the single-cell differential analysis we performed.

How many total differential sites were identified from bulk ATAC-seq and how many of these were found in the single cell ATAC-seq data?

Here is a table that show the total number of peaks called at the NRXN1, NLGN1 and CDH9 loci. Please see the response above for the list of peaks that match between the TCGA's bulk ATAC-seq and sci-ATAC-seq data sets.

	TCGA total peak number at gene locus	Sci-ATAC total peak number at gene locus	Number of matched peaks	Distance (bp)
NRXN1	23 peaks	61 peaks	18 peaks match	0-238 bp apart
NLGN1	23 peaks	35 peaks	No matches	N/A
CDH9	7 peaks	30 peaks	3 matches	124-185 bp apart

Was NRXN1 higher expressed in high-grade compared to low grade tumors (which would be expected based on the higher accessibility detected)? The authors state that NLGN1 was not detected in RNA-seq in TCGA; what about CDH9? And how does it relate to the cyclic IF data?

Based on the bulk RNA-seq data from the TCGA, which included clinical stages overlapping with our patient cohort (Gleason score 4+4), but excluded a major clinical category in our study (Gleason score 3+3), we did not observe a statistically significant higher expression of NRXN1 in high-grade prostate tumors. However, we observe that NRXN1 was expressed at the mRNA level in stage Gleason score 3+4 and higher prostate cancers in the TCGA's cohort of patients (Supplementary Figure 8A). NLGN1 is detected at low levels (Supplementary Figure 8F) and CDH9 is not detected. We changed the results section accordingly and we thank the reviewer for directing us to clarify this point in the new version.

“Next, we analysed the bulk RNA-seq profiles of 497 patients from the TCGA cohort and detect NRXN1 expression in the majority of the prostate tumours (Supplementary Figure 8). In contrast, we detect low levels of NLGN1 transcripts and no CDH9 transcription in the TCGA bulk RNA-seq data set (Supplementary Figure 8).”

Our differential analysis of the sci-ATAC-seq data identified three targets, CDH9, NLGN1 and NRXN1. We were not able to find a commercial antibody for CDH9 to validate the expression of this marker. NLGN1 and NRXN1 are known to physically interact with each other, i.e., they are each other's targets in different biological contexts. Therefore, we decided to pursue the expression of these two neuronal adhesion molecules using tissue-sections.

In our cohort of patients, we observe that both NLGN1 and NRXN1 are expressed at the protein level in Gleason pattern 3 and 4 tumors. The signal intensity levels from immunofluorescent staining with primary and secondary antibodies qualitatively suggest an overall increase in Gleason pattern 4 tumors (Figure 5). We also observe a higher signal intensity in Gleason pattern 4 tumors, when we measure signal intensity levels of NRXN1 expression in the cytoplasm and in the nucleus. However, signal intensity values obtained from a secondary antibody staining across FFPE tissue-sections acquired from different patients are always semi-quantitative because of high background autofluorescence levels of FFPE tissue-sections, the absence of objective methods for thresholding signal levels and the lack of universal antibody signal-intensity normalization methods across patient samples. Therefore, we focused on the quantitative measurement of distinct cell types with NRXN1 and NLGN1 expression in low- and high-risk prostate cancer patients, which show a higher percentage of certain cell populations expressing these neuronal adhesion molecules in high-risk prostate cancer patients (Figure 5).

2) Overall, it is not clear how tumor cell clusters were identified, e.g. how were normal epithelial cells distinguished from cancerous epithelial cells?

We thank the reviewer for this crucial comment. We significantly improved the annotation of our cell clusters based on the comments from both reviewers. Please see the new figure and supplementary figure (Figure 2E, Supplementary Figures 2 and 3) and the new Results subsection added to the manuscript titled: *“Epithelial prostate cancer cells carry markers of inflammatory response”*, also included on pages 5-7 in this document.

The authors state in line 212-214 that Gleason 3 tumors formed without patient-specific clusters and point to Supp Figure 4, but Supp Figure 4 shows 13 clusters (panel B) and individual samples are located to distinct places on the UMAP. Would this not indicate patient-to-patient differences?

We thank the reviewer for this comment and we agree that it is very difficult to see the comparison between patient-based clusters (Supplementary Figure 6A) and topic-based clusters (Supplementary Figure 6B) in this figure. To clearly and objectively show how many cells from each patient sample contribute to each Topic cluster, we generated an additional figure (Supplementary Figure 6C). This new figure shows that, even though samples 5, 11 and 18 dominate three distinct clusters on the UMAP, the rest of the Gleason pattern 3 samples exhibit a uniform distribution across the remaining clusters. More importantly, our new addition to the manuscript, the Silhouette analysis, supports the hypothesis that cells from Gleason pattern 3 tumors are more heterogeneous as compared to cells from Gleason pattern 4 tumors (Figure 3F).

“Supplementary Figure 6C. Distribution of cells from each tumour sample into different clusters.”

ERG motif also seems to be enriched mostly in a subset of cells which shows the heterogeneity within this large cluster.

Yes, we think there is evidence that two tumors in our analysis (one high-grade and one intermediate grade prostate tumor) may have many cells with the ERG fusion based on the snapATAC gene annotation tool results (Supplementary Figure 7A). We think it is interesting that this potential ERG overexpression (based on the accessibility results observed in cells from Samples 4 and 16) did not result in a separate cluster in the UMAP space. We observed that Topics 4 and 8 are enriched in these samples, which returned the top GO terms epithelial cell development (T4) and circadian regulation of gene expression (T8). Some of the genes that define these GO terms include AR, GRHL2, FOXA1, SLC43A1, WNT7B, which are known downstream players active in ERG expressing prostate cancer cells.

Based on this suggestion we revised the manuscript as follows: *“ERG, ETV1, ETV4 and FLI1 display patient-specific patterns even though accessibility of these markers did not drive clustering of cells within the data (Supplementary Figure 7A). Samples 4 (Gleason score 4+4) and 16 (Gleason score 3+4) contain cells with high ERG accessibility, whereas the rest of the samples show heterogenous distribution for ERG accessibility (Supplementary Figure 7A). Topics enriched in Samples 4 and 16 include genes such as AR, GRHL2, FOXA1, SLC43A1, WNT7B, which are known downstream players active in prostate cancers with ERG expression. Sample 3 contain cells with high FLI1 accessibility. ETV1 and ETV4 show a heterogenous distribution (Supplementary Figure 7A).*

Supp Fig 1 B indicates that indeed most of the clusters are dominated by nuclei from one patient.

To address the possibility of Sample_15 dictating clusters identified through the cisTOPIC, we eliminated this sample from our analysis and performed differential analysis between the remaining single-cells from Gleason pattern 3 and 4 tumors. Our results show very similar results in terms of differential accessibility, identifying synaptic adhesion molecules NRXN1, NLGN1 and CDH9 as the top GO term. We added this additional analysis as Supplementary Figure 4B in the revised manuscript. We also provided sample resolved genome browser tracks as a new figure (Supplementary Figure 5). Similarly, when we eliminate Sample_15, in addition to Samples 3 and 4 from our analysis, we observe a similar distribution of Gleason pattern 3 samples on the UMAP (Supplementary Figure 6), providing another line of evidence that even though Sample_15 contributed many cells to our sci-ATAC-seq analysis, it did not skew the biological results we obtained in the study. Additionally, we also provided the sample resolved heatmap as a new panel to Supplementary Figure 3, which shows Topics identified in the paper does not correlate with any individual samples.

** See Nature Research's author and referees' website at www.nature.com/authors for information about policies, services and author benefit

REVIEWER COMMENTS

Reviewer #1 (Remarks to the Author):

If there were an award for most-improved, this manuscript would win. I was honestly skeptical that I would see this again, but I am happy to say that I was wrong. Kudos for being incredibly responsive to the critiques and making this into a worthy story.

I do have a few minor suggestions:

1. The title should reflect the impact which should include something about loss of heterogeneity or identification of genes in Gleason 4 that might promote epi-neuronal interactions.
2. I would hesitate to use the term 'stromal' when you are really referring to 'neurons'. Most people associate stromal with 'fibroblasts and muscle'. Also, the changes in stroma are modest at best, not sure the data support the inclusion of stromal changes as significant.
3. I would highlight more the potential impact of NRXN1 in epi-neuron interaction as the most exciting future direction.
4. The NLGN1 staining looks like it's working in neurons, but not increasing in Gleason 4 tumor epi. The NRXN1 is the only marker that looks like it's really increasing. You can tell that the background for NLGN1 is just higher in Gleason 4. Be very careful here.
5. I still think you should not be using GREAT topic analysis to define your clusters. The clusters should be annotated as discrete cell types through inferred gene expression, annotated and shown in Figure 2. The topic analysis is just confusing and not very important to the conclusion of decreased heterogeneity and increased neuronal markers. Also, it also looks like there is a basal cell cluster according to the KRT14/KRT5 plots in the supp data. Basal cells have been suggested as a pool of progenitors so it would be interesting to single that cluster out and check if the open chromatin is different than in luminals. On that note, please annotate clusters ON the UMAP plot where possible. The colors are very difficult to discriminate from the legend.

Reviewer #2 (Remarks to the Author):

The authors have sufficiently addressed my comments.

However, in my opinion the manuscript would benefit if several figures and figure panels could be reworked. E.g. there are several instances of too low or inconsistent font size, e.g. peak labels in 1C, term labels in 2D, axis in 3E and F, Fig 4 A,B vs 4C; if items cannot be displayed in legible form and don't contribute to the conclusion they could also be deleted e.g. TF labels in Fig. 4; also, figure panel labels should be aligned to the panel and not overlap with panels, e.g. Figure 2. It is also confusing that in figure 3 panel E is to the right of panel F.

Revision #2

REVIEWER COMMENTS

Reviewer #1 (Remarks to the Author):

If there were an award for most-improved, this manuscript would win. I was honestly skeptical that I would see this again, but I am happy to say that I was wrong. Kudos for being incredibly responsive to the critiques and making this into a worthy story.

We genuinely thank the reviewer for helping us significantly improve this manuscript.

I do have a few minor suggestions:

1. The title should reflect the impact which should include something about loss of heterogeneity or identification of genes in Gleason 4 that might promote epi-neuronal interactions.

We agree with the reviewer about the title. We changed the title of our manuscript to: "Epigenetic loss of heterogeneity from low to high grade localized prostate tumors" to better reflect our findings in the paper.

2. I would hesitate to use the term 'stromal' when you are really referring to 'neurons'. Most people associate stromal with 'fibroblasts and muscle'. Also, the changes in stroma are modest at best, not sure the data support the inclusion of stromal changes as significant.

We thank the reviewer for pointing this out. We changed the word "stromal" to "endothelial, immune and neuronal" throughout the text when we are reporting NRXN1 and NLGN1 expression in the prostate tumor microenvironment. We completely agree that the changes in stromal NRXN1 and NLGN1 expression are very modest, but the observation that these molecules are expressed in endothelial, immune and neuronal cells in prostate tumors is novel and requires further investigation.

3. I would highlight more the potential impact of NRXN1 in epi-neuron interaction as the most exciting future direction.

We agree with the reviewer that NRXN1 may have an impact in cancer-neuron interactions. However, we are also very curious to investigate the expression of NLGN1 in tumor innervating neuronal cells more closely in the future.

4. The NLGN1 staining looks like it's working in neurons, but not increasing in Gleason 4 tumor epi. The NRXN1 is the only marker that looks like it's really increasing. You can tell that the background for NLGN1 is just higher in Gleason 4. Be very careful here.

Yes, this really requires a deeper analysis of neuronal and prostate cancer cells in localized prostate tumors. It is very possible that the expression level of NLGN1 does not change in cancer cells, but the chromatin accessibility for the NLGN1 changes in response to cues from the tumor microenvironment. We are very excited to analyze the upstream events that causes these changes in chromatin accessibility in our future work.

5. I still think you should not be using GREAT topic analysis to define your clusters. The clusters should be annotated as discrete cell types through inferred gene expression, annotated and

shown in Figure 2. The topic analysis is just confusing and not very important to the conclusion of decreased heterogeneity and increased neuronal markers.

We agree with the reviewer that including marker gene information that is currently presented in Supp fig 2 is a valuable component of cell type identification and cluster classification. We now include these plots in the revised main Figure 2. We also moved GREAT analysis results and GO terms for immune clusters to the Supplementary Figure 2. However, we found that the topic-based analysis provided the most insight into the specific characterizing properties of the various epithelial clusters and therefore believe it is an important component of Figure 2.

Figure 2E: Gene scores are shown for epithelial cell type markers luminal (KRT8), basal (KRT5 and KRT14) and neuroendocrine (Chromogranin A); common prostate epithelial cell markers

AR, TMPRSS2 NKX3.1; prostate cancer cell markers AMACR, EPCAM, CDH1, C-MYC and PTEN.

Also, it also looks like there is a basal cell cluster according to the KRT14/KRT5 plots in the supp data. Basal cells have been suggested as a pool of progenitors so it would be interesting to single that cluster out and check if the open chromatin is different than in luminals.

We thank the reviewer for their input. We added this observation to our Results section. This is definitely a very interesting point that we will pursue in the future.

On that note, please annotate clusters ON the UMAP plot where possible. The colors are very difficult to discriminate from the legend.

We annotated clusters on the UMAP in the new version.

Reviewer #2 (Remarks to the Author):

The authors have sufficiently addressed my comments.

We very much appreciate the feedback we received from the reviewer, which helped us improve our manuscript.

However, in my opinion the manuscript would benefit if several figures and figure panels could be reworked. E.g. there are several instances of too low or inconsistent font size, e.g. peak labels in 1C, term labels in 2D, axis in 3E and F, Fig 4 A,B vs 4C; if items cannot be displayed in legible form and don't contribute to the conclusion they could also be deleted

We thank the reviewer for this list of specific figure modifications.

peak labels in 1C: this is deleted since these labels do not carry any information.

term labels in 2D: font size has been changed to Arial 18 from Arial 12 and moved on the blue bars for ease of following the list of GO terms.

axis in 3E and F: font size has been changed to Arial 12 from Arial 8.

Fig 4 A,B vs 4C; font size has been changed to Arial 12 from Arial 8.

e.g. TF labels in Fig. 4;

We changed the font size from point 6 to point 10 for all TF labels and deleted the ones that are below the threshold.

also, figure panel labels should be aligned to the panel and not overlap with panels, e.g. Figure 2.

Figure 2 has changed in the new version in response to the comments of Reviewer #1. We made sure to align all figure panel labels for Figure 2 in the new version.

It is also confusing that in figure 3 panel E is to the right of panel F.

Fig 3E is moved to the left and 3F to the right.

Reviewers' Comments:

Reviewer #1:

Remarks to the Author:

The authors have adequately addressed my comments. Congrats to all.

Doug Strand

Reviewer #2:

Remarks to the Author:

The authors have addressed my comments.